# MERGE-OF-THOUGHT DISTILLATION

## ABSTRACT

Efficient reasoning distillation for long chain-of-thought (CoT) models is increasingly constrained by the assumption of a single oracle teacher, despite the practical availability of multiple candidate teachers and growing CoT corpora. We revisit teacher selection and observe that different students have different "best teachers," and even for the same student, the best teacher can vary across datasets. Therefore, to unify multiple teachers' reasoning abilities into a student to overcome conflicts among various teachers' supervision, we propose **Merge-of-Thought Distillation (MoT)**, a lightweight framework that alternates between teacher-specific supervised fine-tuning branches and weight-space merging of the resulting student variants. On competition math benchmarks, using only about 200 CoT samples, applying MoT to a Qwen3-14B student surpasses strong models including Deepseek-R1, Qwen3-32B, and OpenAI-O1, demonstrating substantial gains. Besides, MoT consistently outperforms the best single-teacher distillation, improves general reasoning beyond mathematics while reducing catastrophic forgetting, and shows robustness to distribution-shifted and peer-level teachers. Finally, we have demonstrated MoT possesses consensus CoT by eliminating teacher-specific inductive biases and inter-teacher conflicts while repeatedly reinforcing the learning of consensus reasoning features. These results position MoT as a simple, effective route to efficiently distilling long CoT capabilities from diverse teachers into compact students.

## 1 INTRODUCTION

As large language models (LLMs) with long chain-of-thought (CoT) capabilities continue to emerge (Jaech et al., 2024; Yang et al., 2025a; Guo et al., 2025), reasoning distillation is becoming the key pathway for converting expensive reasoning ability into deployable efficiency. Compared with imitating only final answers, directly supervising the reasoning trajectory enables a smaller student model to learn multi-step solution procedures (Luo et al., 2025b; Qin et al., 2025; Guo et al., 2025).

Building on these developments, the research focus is shifting from scaling data volume to improving data quality. For example, supervised fine-tuning on only 1,000 teacher-distilled samples delivers measurable reasoning gains when paired with test-time compute (Muennighoff et al., 2025). Likewise, when pretraining already imparts rich mathematical knowledge, a few hundred carefully curated examples can effectively elicit complex reasoning (Ye et al., 2025). Taken together, these findings indicate that efficiently distilling long CoT trajectories is an effective strategy for training compact models that achieve competitive reasoning accuracy.

However, real-world deployments rarely features a "single oracle teacher." We often have multiple candidate teacher LLMs and a growing pool of distilled CoT data, giving rise to a basic question: *Given a student model, how we pick the most suitable teacher?* Empirically, teacher choice matters—the teacher can imprint a recognizable "style signature" on the student (Chen et al., 2025b); mismatches between teacher and student can weaken the transfer of long CoT skills (Wu et al., 2025b). As illustrated in Figure 1, our observations are consistent: different students have different "best teachers," and even for the same student the best teacher can vary across datasets. Such phenomena challenge the naive assumption that "a bigger/stronger teacher is necessarily better," prompting us to consider: *Instead of being constrained by a single teacher and the inherent costs of its selection, a more robust and effective paradigm involves aggregating knowledge from multiple teachers.*

A natural follow-up question is: ***How can we effectively fuse the diverse strengths of multiple teachers?*** The goal is to consolidate their complementary reasoning features into a single student. Long CoTs often accumulate noise and irrelevant content (Luo et al., 2025a; Zhang et al., 2025; Li et al., 2025b). It is unclear whether, in mixed-teacher long-CoT distillation, such noise is amplified through interactions, and how to suppress noise while preserving the consensus features. It suggest that: ***Diversity of teachers and reasoning paths is an asset—provided we can overcome conflicts among the supervision of various teachers***.

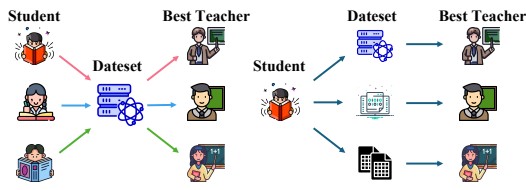

Figure 1: Teacher choice is not universal. Left: different students have different "best teachers"; right: even for the same student the best teacher can vary across datasets. This observation is empirically confirmed in Table 1.

As an effective technique for overcoming data distribution conflicts, model merging has been widely applied to joint training across diverse domains and tasks (Yu et al., 2024b; Zhou et al., 2024; Yadav et al., 2024). However, our revisiting analysis also showed that a single Post-hoc merge does not reliably resolve cross-teacher supervision conflicts and unify different teachers' reasoning abilities. These limitations motivate an approach that goes beyond one-shot merging to reconcile heterogeneous teacher signals by **repeatedly reinforcing the learning of consensus reasoning features**.

To this end, we propose **Merge-of-Thought Distillation (MoT)**: a lightweight framework that alternates between (i) **teacher-specific branch SFT** and (ii) **weight-space merging of student variants**. Intuitively, branch SFT internalizes the reasoning style of each teacher into one student; the subsequent parameter-space merge then distills consensus—retaining features reinforced across teachers while suppressing individual accidents and quirks. After multiple iterations, the student progressively condenses into a merged student that reflects multi-teacher consensus reasoning. We found that MoT significantly enhanced reasoning ability of the model and alleviated catastrophic forgetting. In addition, we have experimentally and theoretically demonstrated that **consensus CoT emerges naturally with MoT**: MoT eliminates teacher-specific inductive biases and inter-teacher conflicts at the token level while repeatedly reinforcing the learning of consensus reasoning features, enabling training in a flatter loss landscape and effective transfer to new student models.

We present, to our knowledge, the **first systematic study of multi-teacher long CoT co-distillation**:

1. We conduct the revisiting analysis of teacher selection under Long CoT distillation setting and find that there is no single best teacher consistently dominant across students or datasets.

2. Rather than taking the cost on teacher selection, we propose a novel distillation method, **Merge-of-Thought Distillation (MoT)**, to unify multiple teachers' reasoning abilities into students by overcoming conflicts among the supervision of various teachers.

3. Using only about 200 CoT samples, applying MoT to a Qwen3-14B student surpasses strong models including Deepseek-R1, Qwen3-32B, and OpenAI-O1. Besides, MoT consistently outperforms the best single-teacher distillation, improves general reasoning beyond mathematics while reducing catastrophic forgetting, and shows robustness to distribution-shifted and peer-level teachers.

4. We have demonstrated MoT possesses consensus CoT by eliminating teacher-specific inductive biases and inter-teacher conflicts while repeatedly reinforcing the learning of consensus reasoning features, which enables the model to be trained on a flatter loss landscape and further propagated to new student models.

## 2 RELATED WORK

**Long Chain-of-Thought Distillation.** Research on distilling long chains of thought (CoT) has progressed rapidly (Wu et al., 2025b; Guo et al., 2025). Early work (Li et al., 2023) showed that even small models can benefit from teacher CoT prompting and highlighted the importance of varied reasoning chains. Subsequent approaches (Luo et al., 2025b; Feng et al., 2024) further segment

and simplify CoTs, employ keypoint weighting, and use progressive distillation to focus on critical tokens. Studies on the key factors of CoT distillation reveal that teacher diversity and rationale granularity often have a greater impact than raw teacher accuracy (Chen et al., 2025b). Recent works also show that long-CoT capability can be bootstrapped with a handful of in-context examples (Pang et al., 2025), distilled as summaries to improve long-context memory (Ma et al., 2025), or integrated with vision reasoning using agent-based approaches (Shi et al., 2024). These findings underscore that long-CoT distillation not only requires carefully curated examples but also faces challenges such as **teacher selection**, **noise amplification** and **distillation efficiency**. Nevertheless, most existing methods focus on a **single teacher distillation**; our work instead extends this line of work by fusing multiple teachers' reasoning abilities into a single student to achieve stronger performance.

**Model Merging in LLMs.** Model merging fuses the parameters of multiple trained models into a single model, which is distinct from output-level ensembles (Yang et al., 2024; Tam et al., 2024). Empirical studies show that merging tends to balance performance and safety better than mixing data across tasks or languages (Yang et al., 2025b; Yadav et al., 2024; Yu et al., 2024b; Jin et al.). More advanced techniques adapt merging to pre-trained models by disentangling weights into magnitude and direction (Yu et al., 2024a). Other approaches merge checkpoints during pre-training for faster convergence or use activation importance to retain critical parameters (Li et al., 2025a; Nobari et al., 2025). Model merging has also been applied to combine models with different reasoning strategies and to merge heterogeneous architectures (Wu et al., 2025a; Zhang et al., 2024). However, most existing work focuses on merging models specialised for different domains and tasks; by contrast, our approach merges student models distilled by different teachers on the same dataset to **unify their reasoning abilities** without conflicts among different teachers.

Table 1: Best teacher under STD for each base model and dataset.

| Base model | Best teacher on BOBA-200 | Best teacher on S1K-200 |
|---|---|---|
| Qwen3-8B | QWQ | QWQ |
| Qwen3-14B | Qwen3-235B | QWQ |
| Qwen3-30B-A3B | Qwen3-235B | Qwen3-235B |

## 3 REVISITING MULTI-TEACHER LONG CoT DISTILLATION

**Setup and goals.** We fine-tune three students from the Qwen3 family (Qwen3-8B / Qwen3-14B / Qwen3-30B-A3B) on two teacher-distilled math subsets, BOBA-200 and S1K-200. We compare three regimes: (i) **single-teacher distillation (STD)**, (ii) a **direct multi-teacher union (MTD)** that mixes all available teacher-distilled samples, and (iii) a **one-shot post-hoc weight merge** of students independently distilled from different teachers. Further dataset/model/training details appear in Sec. 5. This section has two goals: (1) revisit teacher selection under long CoT distillation and quantify that the best teacher is student and dataset-dependent; and (2) show that naive MTD or a single *post-hoc* merge does not reliably resolve cross-teacher supervision conflicts.

Table 2: Final AIME24/25 *AVG* under three regimes. MTD denotes naive multi-teacher union. Best STD denotes best single-teacher for that setting. MTD and Post-hoc weight merge do not reliably overcome cross-teacher conflicts or unify heterogeneous reasoning styles.

| Base Model | Dataset | Baseline | MTD | Best STD | Post-hoc Merge |
|---|---|---|---|---|---|
| Qwen3-8B | BOBA-200 | 71.46 | 72.50 | 71.88 | 73.12 |
| | S1K-200 | 71.46 | 73.23 | 72.09 | 73.02 |
| Qwen3-14B | BOBA-200 | 74.59 | 75.94 | 76.98 | 76.98 |
| | S1K-200 | 74.59 | 76.26 | 76.57 | 76.26 |
| Qwen3-30B-A3B | BOBA-200 | 75.77 | 76.67 | 78.65 | 78.54 |
| | S1K-200 | 75.77 | 76.46 | 77.61 | 77.08 |

**Different students have different best teachers.** Table 1 summarizes, which single-teacher distillation (STD) source achieves the best distillation performance for each base model and dataset.

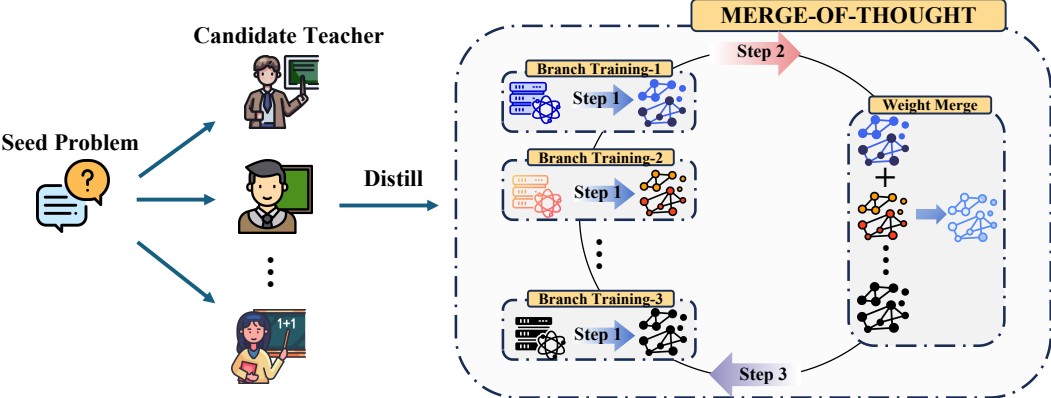

Figure 2: Workflow of Merge-of-Thought Distillation (MoT). After the candidate teachers generate the teacher-specific distillation dataset based on the seed problem, the system enters the iterative MoT algorithm process. In each round $t$, we perform three steps: **Step 1 (branch training)**: initialize $K$ branches from the current merged student and train each on its teacher-specific distillation dataset $\mathcal{D}^{(k)}$ (Eq. 1); **Step 2 (weight merge)**: average the branch parameters in weight space to obtain the aggregated model $\theta^{(t)}$ (Eq. 2); **Step 3 (next-round initialization)**: use $\theta^{(t)}$ as the base initialization for round $t+1$.

We observe that different students have different best teacher, and **even for the same student the best teacher can vary across datasets**. This revisiting analysis challenges the naive belief that a larger/stronger teacher is always better. Details are provided in the Table 4.

**Simple mixing or one-shot post-hoc merging is insufficient.** Table 2 reports final AIME24/25 AVG across two datasets and three student scales. While MTD often improves over the base model, it sometimes lags behind the best per-setting STD especially when the scale of the student model grows. In practice, post-hoc merging behaves similarly to MTD. This means that a straightforward MTD that directly unioning all teachers' distilled samples, and a single post-hoc weight merge of independently distilled students **do not reliably overcome cross-teacher conflicts or unify heterogeneous reasoning styles, motivating the need for an iterative merge-and-train approach** introduced next in Sec. 4 to reconcile heterogeneous teacher signals by repeatedly reinforcing the learning of consensus reasoning features.

# 4 METHOD: MERGE-OF-THOUGHT DISTILLATION (MOT)

Our approach assumes access to a base language model, a small set of supervised problems with reference answers, and multiple teacher models. The core idea is to consolidate reasoning signals that are consistent across heterogeneous teacher rationales. MoT alternates between teacher-specific supervised fine-tuning (SFT) branches and weight-space merging, and is performed iteratively. Concretely, MoT consists of two core steps repeated for multiple rounds:

1. *Branch training (teacher-specific SFT):* For each teacher, fine-tune a branch of the student on that teacher's rationales.

2. *Weight merge:* Merge branch parameters by averaging to form the next student initialization.

We detail the setup and these steps below. An overview of the approach is illustrated in Figure 2.

## 4.1 INITIALIZATION

**Data.** Let $\mathcal{D} = \{(x_i, y_i)\}_{i=1}^N$ be a set of problems $x$ with reference answers $y$. We consider $K$ teacher models. For each input $x$, teacher $\tau_k$ produces a rationale $r^{(k)}$ and a final answer $\hat{y}^{(k)}$. When $y$ is available, we optionally retain only the teacher outputs that match the reference answer, yielding teacher-specific datasets:

$$\mathcal{D}^{(k)} = \{(x_i, r_i^{(k)})\}_{i=1}^{N_k},$$

which filters out teacher trajectories that do not reach the correct final answer.

**Model.** Let $m$ denote the student with parameters $\theta$. We initialize from the base model parameters $\theta^{(0)}$ and iterate the MoT procedure for $t = 1, \ldots, T$ rounds.

## 4.2 TEACHER-SPECIFIC SFT (BRANCH TRAINING)

**Targets.** For each teacher $k$, we train the student to produce the teacher's rationale:

$$\text{target}(x; k) = r^{(k)}.$$

This choice encourages the student to internalize teacher-specific reasoning patterns, rather than only the short final answer.

**Objective.** The SFT objective for teacher $k$ is the token-level cross-entropy over the target sequence:

$$\mathcal{L}_{\text{SFT}}^{(k)}(\theta) = \mathbb{E}_{(x,r^{(k)},y)\sim\mathcal{D}^{(k)}} \sum_{t=1}^{L(x,k)} -\log p_\theta\big(z_t \mid x, z_{<t}\big), \tag{1}$$

where $z_{1:L(x,k)}$ tokenizes $\text{target}(x; k)$. In round $t$, we initialize $K$ branches from the current merged model and fine-tune each branch on its teacher's data:

$$\theta^{(t,k)} \leftarrow \arg\min_\theta \; \mathcal{L}_{\text{SFT}}^{(k)}(\theta) \quad \text{with init } \theta^{(t-1)}.$$

## 4.3 WEIGHT-SPACE MERGING AND ITERATION

After branch training, we merge the $K$ branch parameters by averaging to get the next initialization:

$$\theta^{(t)} = \frac{1}{K} \sum_{k=1}^{K} \theta^{(t,k)}. \tag{2}$$

This step consolidates reasoning features that are shared across branches while smoothing out teacher-specific noises. We repeat the two steps—branch training and weight merge—for $T$ rounds, resulting in the final merged model $\theta^{(T)}$. We aim to leverage *model merging* to overcome conflicts among various teachers' supervision and, through continuous merge-and-training iterations, unify different teachers' reasoning abilities and ultimately converge to a consensus reasoning landscape.

## 5 EXPERIMENTS SETUP

**Datasets.** We work in a one-question–multiple-answers (1Q–multiA) setting. We use two high-quality open-source mathematical datasets (BOBA (inclusionAI, 2025) and S1K (Muennighoff et al., 2025) as our source datasets. From each source dataset, we sample 200 prompts and denote the resulting subsets as BOBA-200 and S1K-200. For every prompt, we query four teacher models—Qwen3-32B (Yang et al., 2025a), QWQ (Team, 2024b), Deepseek-R1 (Guo et al., 2025), and Qwen3-235B (Yang et al., 2025a). Each teacher generates 16 responses with temperature set to 0.6 and max_tokens set to 32,768. For distillation, we randomly select one correct reasoning path among the 16 as the training label; if none of the 16 responses is correct, we discard that prompt for the corresponding teacher's distillation corpus. We construct two training regimes:
(1) **Single-Teacher Distillation (STD)**, where we build one distilled corpus per teacher.
(2) **Multi-Teacher Distillation (MTD)**, where we aggregate all available distilled samples from all teachers for each source.
The resulting STD and MTD datasets and their sizes are summarized in Table 21. Rows with a specific teacher correspond to STD, while rows with "ALL TEACHERS" correspond to MTD.

**Sampling strategy for BOBA-200 and S1K-200.** Following the general observation that random sampling can lead to variable prompt difficulty in reasoning tasks (Wang et al., 2025b), we adopt a simple but reproducible sampling strategy for our subsets. For BOBA-200, we directly use the default 200 problems provided by the official BOBA release, which are themselves obtained by random sampling from the full benchmark, without any additional filtering or manual selection. For S1K-200, since some items are proof-style questions (about 200 items) without a verifiable final answer, we first remove all such problems and then uniformly sample 200 prompts at random from

the remaining questions that have a boxed checkable answer. In both cases, we keep the process as random as possible under the constraint of automatic answer verification and use exactly the same batch of prompts for all comparative experiments (MoT, all STDs, and MTD) to avoid cherry-picking and minimize sensitivity to a particular sample.

**Training Configuration.** We fine-tune Qwen3-8B, Qwen3-14B, and Qwen3-30-A3B (Yang et al., 2025a) as base models across all experiments. For MoT, the base model alternates training on each of the four STD corpora for 50 steps and then performs a merge; this constitutes one merge round. We run 5 merge rounds in total and report the best-performing round as the final MoT result; For STD and MTD baselines, to ensure fairness, we train for 250 steps in total and save a checkpoint every 50 steps. We also report the best-performing checkpoint as the final result. More details are provided in the Appendix G. We evaluate the capabilities of the model in mathematical reasoning using AIME24 (Math-AI, 2024) and AIME25 (Math-AI, 2025). All AIME scores are 16-run averages.

Table 3: Main results with MoT on BOBA-200 and S1K-200. "/" denotes an item not reported in the corresponding baseline's source. All AIME scores are 16-run averages.

| Base Model | Configuration | Annotated Examples | AIME24 | AIME25 | AVG | AVG Gain |
|---|---|---|---|---|---|---|
| | Base | — | 75.83 | 67.08 | 71.46 | - |
| Qwen3-8B | DEER (Dai et al., 2025) | 103K | 76.70 | / | - | - |
| | S-GRPO (Dai et al., 2025) | 103K | 77.30 | / | - | - |
| | MathSmith-HC (Zhan et al., 2025) | 11K | 76.70 | 70.00 | 73.35 | ↑1.89 |
| | **BOBA-200 + MoT (Ours)** | 200 | 78.33 | 70.63 | 74.48 | ↑3.02 |
| | **S1K-200 + MoT (Ours)** | 200 | 77.50 | 71.67 | 74.59 | ↑3.13 |
| | Base | — | 13.75 | 11.46 | 12.61 | - |
| QWEN2.5-14B | GRPO (Chen et al., 2025a) | 1K | 13.33 | 13.13 | 13.23 | ↑0.62 |
| | SPO (Chen et al., 2025a) | 1K | 14.17 | 16.67 | 15.42 | ↑2.81 |
| | RefCritic (SFT) (Tang et al., 2025) | 10K | 15.20 | 15.00 | 15.10 | ↑2.49 |
| | RefCritic (SFT+RL) (Tang et al., 2025) | 120K | 23.00 | 21.20 | 22.10 | ↑9.49 |
| | Bespoke-Stratos-17k (Kou et al., 2025) | 17K | 20.00 | 13.30 | 16.65 | ↑4.04 |
| | Difficulty-Flipped (Kou et al., 2025) | 17K | 23.00 | 23.30 | 23.15 | ↑10.54 |
| | Long-CoT (Wang et al., 2025a) | 220K | 30.00 | / | - | - |
| | **BOBA-200 + MoT (Ours)** | 200 | 34.17 | 30.00 | 32.09 | ↑19.48 |
| | **S1K-200 + MoT (Ours)** | 200 | 36.88 | 30.42 | 33.65 | ↑21.04 |
| Qwen3-14B | Base | — | 79.17 | 70.00 | 74.59 | - |
| | **BOBA-200 + MoT (Ours)** | 200 | 79.38 | 76.88 | 78.13 | ↑3.54 |
| | **S1K-200 + MoT (Ours)** | 200 | 81.67 | 75.63 | 78.65 | ↑4.06 |
| Qwen3-30B-A3B | Base | — | 80.63 | 70.90 | 75.77 | - |
| | UloRL-A3B-32k (Du et al., 2025a) | / | / | 73.50 | - | - |
| | **S1K-200 + MoT (Ours)** | 200 | 80.83 | 77.50 | 79.17 | ↑3.40 |
| | **BOBA-200 + MoT (Ours)** | 200 | 82.92 | 78.33 | 80.63 | ↑4.86 |
| Qwen3-32B | Base | — | 81.46 | 72.08 | 76.77 | - |
| Deepseek-R1 | Base | — | 79.80 | 70.00 | 74.90 | - |
| OpenAI-O1 | Base | — | 74.30 | 79.20 | 76.75 | - |
| OpenAI-O3-MINI | Base | — | 79.60 | 74.80 | 77.20 | - |

## 6 MULTI-TEACHER DISTILLATION AND MOT YIELD SUBSTANTIAL GAINS

### 6.1 PERFORMANCE ON COMPETITION MATH BENCHMARKS

**Main results.** To demonstrate the superiority of MoT, we report gains across multiple model scales and compare them against two axes of baselines: (i) larger models like Deepseek-R1, Qwen3-32B and (ii) same-base alternatives trained on methods using substantially larger, differently sourced reasoning datasets. Because the Qwen3 family is very frontier and lacks extensive public baselines, we additionally include results of applying MoT to Qwen2.5-Instruct-14B (Team, 2024a) as a complementary case to test the effectiveness of MoT on 14B scale.

Table 3 reports the final results of MoT on BOBA-200 and S1K-200. For example, "Qwen3-8B+BOBA-200" denotes Qwen3-8B trained with MoT on BOBA-200 dataset. As shown, **with only 200 training examples** from either BOBA-200 or S1K-200, MoT lifts Qwen3-8B to match

the baseline performance of Qwen3-14B. Moreover, MoT on Qwen3-14B surpasses strong models including Deepseek-R1, Qwen3-32B, and OpenAI-O1, demonstrating substantial gains. In addition, on Qwen2.5-Instruct-14B, MoT's improvements far exceed baselines trained on very large reasoning datasets, reinforcing our claim that **multi-teacher, consensus-based efficient distillation of long CoT reasoning can yield very substantial performance gains.**

**Comprehensive Ablations of STD, MTD, and MoT.** To validate the effectiveness of MoT and multi-teacher distillation, we conduct fine-grained ablations: (1) **STD**: train on each single-teacher distilled dataset (QWQ, Qwen3-32B, Qwen3-235B, Deepseek-R1). (2) **MTD**: train on the union of all teachers' distilled samples. (3) **MoT**: our method that alternates across the four STD corpora with periodic merges. For fairness, all methods save a checkpoint every 50 steps, and we report the best checkpoint; full per-step results are provided in the Appendix G.2.

Table 4: Ablation on STD, MTD, and MoT across settings. AIME scores are 16-run averages.

| Dataset | Method | Qwen3-8B | | | Qwen3-14B | | | Qwen3-30B-A3B | | |
|---------|--------|----------|--------|--------|-----------|--------|--------|---------------|--------|--------|
| | | AIME24 | AIME25 | AVG | AIME24 | AIME25 | AVG | AIME24 | AIME25 | AVG |
| | Baseline | 75.83 | 67.08 | 71.46 | 79.17 | 70.00 | 74.59 | 80.63 | 70.90 | 75.77 |
| | MTD (All Teachers) | 76.04 | 68.96 | 72.50 | 76.46 | 75.42 | 75.94 | 79.38 | 73.96 | 76.67 |
| | STD (QWQ) | 76.25 | 67.50 | 71.88 | **79.58** | 73.54 | 76.56 | 79.79 | 75.63 | 77.71 |
| BOBA | STD (Qwen3-32B) | 75.42 | 67.71 | 71.57 | 77.71 | 71.25 | 74.48 | 81.04 | 76.04 | 78.54 |
| | STD (Qwen3-235B) | 74.58 | 67.92 | 71.25 | 79.17 | 74.79 | 76.98 | 81.88 | 75.42 | 78.65 |
| | STD (Deepseek-R1) | 67.71 | 60.21 | 63.96 | 74.38 | 67.50 | 70.94 | 78.33 | 68.96 | 73.65 |
| | MoT (ours) | **78.33** | **70.63** | **74.48** | 79.38 | **76.88** | **78.13** | **82.92** | **78.33** | **80.63** |
| | Baseline | 75.83 | 67.08 | 71.46 | 79.17 | 70.00 | 74.59 | 80.63 | 70.90 | 75.77 |
| | MTD (All Teachers) | 75.63 | 70.83 | 73.23 | 79.17 | 73.34 | 76.26 | 78.33 | 74.58 | 76.46 |
| | STD (QWQ) | 76.04 | 68.13 | 72.09 | 80.21 | 72.92 | 76.57 | **81.46** | 72.92 | 77.19 |
| S1K | STD (Qwen3-32B) | **77.50** | 66.67 | 72.09 | 79.79 | 72.50 | 76.15 | 79.58 | 73.13 | 76.36 |
| | STD (Qwen3-235B) | 74.38 | 68.54 | 71.46 | 77.08 | 75.41 | 76.25 | 79.17 | 76.04 | 77.61 |
| | STD (Deepseek-R1) | 70.00 | 61.46 | 65.73 | 73.75 | 62.92 | 68.34 | 78.54 | 70.63 | 74.59 |
| | MoT (ours) | **77.50** | **71.67** | **74.59** | **81.67** | **75.63** | **78.65** | 80.83 | **77.50** | **79.17** |

Results are shown in Table 4. MoT consistently yields the strongest distillation gains in almost all settings, which means that MoT is always superior to the optimal result of the teacher selection method under each setting. This indicates that MoT can sidesteps brittle manual teacher selection by fusing complementary reasoning abilities into a single student.

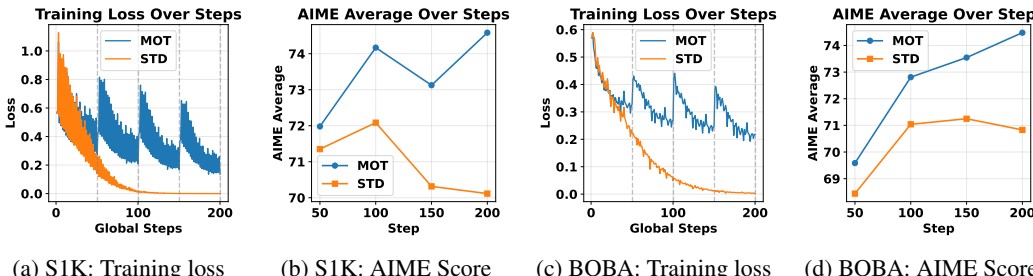

| (a) S1K: Training loss | (b) S1K: AIME Score | (c) BOBA: Training loss | (d) BOBA: AIME Score |

Figure 3: Qwen3-8B under MoT vs. STD (QWQ) on *S1K* and *BOBA*. Panels (a,b): S1K; panels (c,d): BOBA. Left columns show training loss vs. steps; right columns show AIME vs. steps. All runs log loss at every step on the same QWQ-distilled corpus; AIME is evaluated every 50 steps.

**Training Dynamics: MoT vs. Best STD.** We compare Qwen3-8B under MoT and under STD with the best single teacher (QWQ) on both the *S1K* and *BOBA* datasets. We log training loss on the same QWQ-distilled corpora at every step and evaluate AIME score every 50 step. From Figure 3, we observe that MoT achieves substantially higher AIME scores even when its training loss remains much higher than STD's at the same step. This suggests that in long CoT training, lower loss is not necessarily correlated with stronger reasoning ability. Moreover, MoT **exhibits a higher performance ceiling and suppresses overfitting**, with STD typically peaking earlier and then degrading while MoT remains stable or continues improving as steps increase.

## 6.2 COMPUTE–PERFORMANCE TRADE-OFF.

There is an inherent trade-off between computational cost and performance in our setting. For the main BOBA-200 experiments with Qwen3-8B, the training budgets of STD and MoT can be

Table 5: Training budgets of STD and MoT on BOBA-200 with Qwen3-8B.

| Method | # branches | steps / branch | rounds | total branch-steps |
|--------|-----------|----------------|--------|-------------------|
| STD | 1 | 250 | 1 | 250 |
| **MoT** | 4 | 50 | 5 | $4 \times 50 \times 5 = 1000$ |

Table 6: Equal-compute comparison between MoT and RFT-style single-teacher STD on AIME24/25 with Qwen3-8B and BOBA-200. All STD variants are trained with 4 CoTs per question for 1000 steps, matching MoT's total branch-steps and distinct CoT budget.

| Method | AIME24 | AIME25 | AVG |
|--------|--------|--------|-----|
| Base | 75.83 | 67.08 | 71.46 |
| 4×STD (32B) | 75.21 | 67.92 | 71.57 |
| 4×STD (QWQ) | 76.46 | 69.17 | 72.82 |
| 4×STD (235B) | 75.83 | 68.13 | 71.98 |
| 4×STD (R1) | 70.42 | 59.58 | 65.14 |
| **MoT (ours)** | **78.33** | **70.63** | **74.48** |

summarized as shown in Table 5. However, MoT is designed for the realistic setting where multiple teachers are available and one wishes to maximally leverage them rather than commit to a single teacher. In practice, MoT remains highly efficient: it takes only about 6 GPU hours to reproduce the Qwen3+BOBA-200 training on a single 8×H800 machine, and this can be further accelerated by training branches in parallel.

To directly assess whether MoT's gains come purely from increased compute, we ran an additional experiment following the RFT-style setup (Yuan et al., 2023). For each teacher, we re-distilled the data by sampling 4 diverse CoT trajectories per question with high temperature, yielding $4 \times 200 = 800$ rationales per teacher. This matches MoT's total number of distinct CoT sequences (4 teachers × 200 questions × 1 CoT each = 800), so the comparison controls for both the total compute and the amount of distinct CoT supervision. We then performed single-teacher STD for 1000 steps on the chosen teacher's 800 CoTs, matching the total branch-steps of MoT:
**(1) MoT**: 4 branches × 50 steps/branch × 5 rounds = **1000** branch-steps,
**(2) RFT-style STD**: single teacher, 4 CoTs per question (800 CoTs total), **1000** steps.

We saved a checkpoint every 200 steps and report the best checkpoint. The results on AIME24/25 (Qwen3-8B, BOBA-200) are summarized in Table 6.

The strongest single-teacher STD configuration in this equal-compute regime remains competitive, but even with this **strictly matched compute and data budget**, it still underperforms MoT. This indicates that MoT's gains do not arise merely from using more optimization steps; instead, they come from jointly leveraging multiple teachers, avoiding brittle teacher selection, and unifying complementary reasoning signals into a single student, thereby raising the overall reasoning ceiling.

### 6.3 MoT MITIGATES FORGETTING AND STRENGTHENS GENERAL REASONING

To assess whether CoT-style training with MoT affects basic capabilities, we evaluate the final checkpoints trained by MoT and by STD with the per-setting best teacher (Best STD) against the Base models on nine benchmarks: CEVAL (CEV) (Seifert et al., 2024), SUPER_GPQA (SG) (Du et al., 2025b), SIMPLE_QA (SQ) (Wei et al., 2024), IFEVAL (IFE) (Zhou et al., 2023), MMLU_PRO (MP) (Wang et al., 2024), MMLU_REDUX (MR) (Gema et al., 2025), PhyBench (PB) (Meng et al., 2024), LiveCodeBench (LCB) (Jain et al., 2024), and GPQA-Diamond (GPQA-D) (Rein et al., 2024). We group these benchmarks into three categories: **catastrophic-forgetting–sensitive tasks**, **reasoning–knowledge tasks** and **pure reasoning tasks**. Detailed descriptions of these tasks and MoTivations for using and classifying them for evaluation are provided in the Appendix H.

For each configuration, we report raw scores and summarize the average change versus the Base model within each group: "Avg drop" for catastrophic-forgetting tasks and "Avg gain" for reasoning-knowledge and pure reasoning tasks. We report the results in Table 7. Compared with training on the single best teacher, MoT typically yields larger gains on reasoning-knowledge and pure reasoning

tasks while incurring smaller declines on catastrophic-forgetting–sensitive tasks. This suggests that MoT not only **strengthens general reasoning** but also helps **mitigate catastrophic forgetting**. In Appendix E, we provide a more detailed evaluation.

Table 7: Impact of Best STD and MoT on general benchmarks. All scores are 16-run averages.

| Dataset | Base | Config | Catastrophic-forgetting–sensitive tasks | | | | Reasoning-knowledge tasks | | | | Pure reasoning tasks | | | |
|---|---|---|---|---|---|---|---|---|---|---|---|---|---|---|
| | | | CEV | SG | IFE | Avg drop | SQ | MP | MR | Avg gain | PB | LCB | GPQA-D | Avg gain |
| BOBA | 8B | Base | 83.58 | 10.51 | 83.60 | - | 32.31 | 71.42 | 83.21 | - | 20.47 | 55.76 | 57.77 | - |
| | | Best STD | 83.43 | 9.97 | 81.62 | ↓-0.89 | 33.88 | 72.00 | 83.68 | ↑0.87 | 22.85 | 59.88 | 59.85 | ↑2.86 |
| | | MoT | 83.73 | 10.09 | 82.04 | ↓-0.61 | 34.44 | 73.30 | 84.42 | ↑1.74 | 24.07 | 58.79 | 60.54 | ↑3.13 |
| S1K | 8B | Base | 83.58 | 10.51 | 83.60 | - | 32.31 | 71.42 | 83.21 | - | 20.47 | 55.76 | 57.77 | - |
| | | Best STD | 83.95 | 10.18 | 82.35 | ↓-0.40 | 32.75 | 72.24 | 85.02 | ↑1.02 | 22.76 | 59.47 | 56.31 | ↑1.51 |
| | | MoT | 84.32 | 10.15 | 83.51 | ↑0.10 | 33.56 | 73.01 | 84.95 | ↑1.53 | 23.37 | 59.58 | 59.53 | ↑2.83 |
| BOBA | 14B | Base | 86.78 | 10.76 | 84.69 | - | 32.61 | 75.26 | 85.74 | - | 28.53 | 61.41 | 60.83 | - |
| | | Best STD | 83.73 | 10.26 | 82.56 | ↓-1.89 | 32.17 | 74.71 | 86.37 | ↓-0.12 | 30.61 | 63.21 | 63.79 | ↑2.28 |
| | | MoT | 86.70 | 10.38 | 83.51 | ↓-0.55 | 32.65 | 75.59 | 86.53 | ↑0.39 | 30.77 | 63.59 | 64.26 | ↑2.62 |
| S1K | 14B | Base | 86.78 | 10.76 | 84.69 | - | 32.61 | 75.26 | 85.74 | - | 28.53 | 61.41 | 60.83 | - |
| | | Best STD | 84.25 | 10.00 | 84.32 | ↓-1.22 | 32.49 | 76.21 | 86.47 | ↑0.52 | 30.41 | 63.10 | 63.70 | ↑2.15 |
| | | MoT | 85.66 | 10.45 | 84.42 | ↓-0.57 | 32.56 | 76.55 | 86.68 | ↑0.73 | 30.78 | 64.15 | 64.11 | ↑2.76 |
| BOBA | 30B | Base | 85.88 | 10.66 | 83.76 | - | 31.68 | 75.26 | 85.81 | - | 28.57 | 61.08 | 59.76 | - |
| | | Best STD | 84.18 | 10.02 | 80.44 | ↓-1.89 | 31.52 | 75.96 | 86.04 | ↑0.26 | 33.31 | 61.34 | 61.81 | ↑2.35 |
| | | MoT | 86.55 | 10.52 | 83.54 | ↑0.10 | 32.26 | 76.21 | 86.74 | ↑0.82 | 33.46 | 62.54 | 62.34 | ↑2.98 |
| S1K | 30B | Base | 85.88 | 10.66 | 83.76 | - | 31.68 | 75.26 | 85.81 | - | 28.57 | 61.08 | 59.76 | - |
| | | Best STD | 84.62 | 10.04 | 79.74 | ↓-1.97 | 32.40 | 75.49 | 86.67 | ↑0.60 | 33.38 | 63.96 | 61.46 | ↑3.13 |
| | | MoT | 86.48 | 10.14 | 82.91 | ↓-0.26 | 33.19 | 76.49 | 87.28 | ↑1.40 | 33.40 | 63.92 | 62.53 | ↑3.48 |

## 7 MoT ENABLES SELECTION-FREE CoT DISTILLATION

**Ablating a Distribution-Shifted Teacher from MoT: Evidence of Complementarity.** As shown in Table 4, using Deepseek-R1 (R1) as the sole teacher (STD) induces notable performance drops for QWEN bases, indicating a strong distribution shift. To verify that MoT can still leverage useful signals from R1 despite the shift, we ablate R1 from the MoT teacher pool and keep all other settings identical. As shown in Table 8, removing R1 reduces the final MoT performance on BOBA-200 (negative changes), implying that including R1 provides complementary, beneficial supervision that MoT can harness. This proves that MoT can overcome the **performance degradation** caused by the strong distribution shift teacher and extract **beneficial common reasoning features** from it. More details are provided in the Appendix G.3.

**Optimization Dynamics with Distribution-Shifted Teacher.** We visualize optimization dynamics on *BOBA* for both **8B** and **14B** scales under standard MoT and MoT without R1 (removing the R1 teacher). We log training loss at every step on the same QWQ-distilled corpus and evaluate AIME score every 50 steps (as in our ablation protocol). From Figure 4, we observe that although the performance of the no-R1 variants converges faster, including R1 **raises the performance ceiling, delays satu-**

Table 8: Impact of removing R1 from the MoT teacher pool on BOBA-200.

| Base model | AVG change |
|---|---|
| Qwen3-8B | -0.62 |
| Qwen3-14B | -0.21 |
| Qwen3-30B-A3B | -0.42 |

**ration and reduces post-peak degradation**, suggesting better regularization and a higher training upper bound at both scales. This indicates that even with the distribution-shifted teacher, MoT extracts beneficial common reasoning signals while mitigating teacher-specific noise.

**Can peer-level models act as teachers?** We find that teacher usefulness extends beyond strictly stronger models: distilling Qwen3-30B-A3B from peer-level QWQ or Qwen3-32B improves performance. Combining peer-level trajectories with MoT boosts results further (Appendix G.4).

## 8 CONSENSUS CoT EMERGES NATURALLY WITH MoT

**Better student is a better teacher.** To verify that MoT learns higher-quality and more generalizable chains-of-thought (CoT), we conduct a student-as-teacher experiment. Specifically, we take models trained on BOBA-200 under three regimes (Base, Best STD and MoT) and use each **as a teacher** to re-distill on BOBA-200 for a new student model. As shown in Appendix C, when the teacher itself is a student trained with MoT, it almost always provides the **strongest distillation signal**, yielding the best downstream student performance. These results indicate that **consensus CoT**

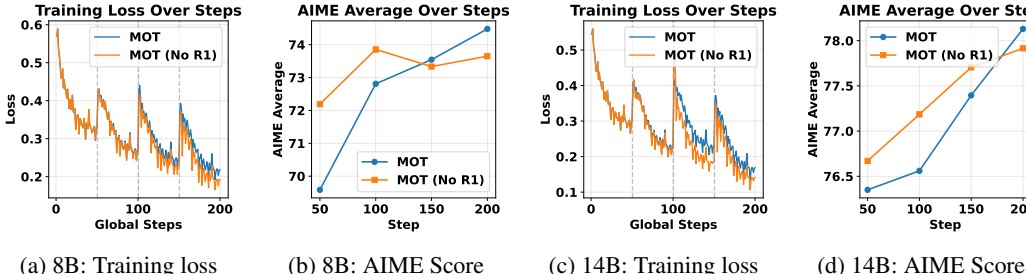

| (a) 8B: Training loss | (b) 8B: AIME Score | (c) 14B: Training loss | (d) 14B: AIME Score |

Figure 4: BOBA dataset: MoT vs. MoT without R1 at two scales. Panels (a,b): 8B; panels (c,d): 14B. Left columns show training loss vs. steps; right columns show AIME vs. steps. All runs log loss at every step on the same QWQ-distilled corpus; AIME is evaluated every 50 steps.

**emerges naturally with MoT**: the student learns trajectories that are both stronger and more consistent, and when used as a teacher, this consensus supervision **transfers** effectively to new students.

**Token-level evidence for consensus CoT.** We further probe token-level confidence on QWQ-distilled CoTs. We mark tokens for which the MoT model's output confidence drops relative to the Base under QWQ teacher's distilled supervision (Figs. 5). Strikingly, the marked tokens concentrate on teacher-specific stylistic expressions (driven discourse markers, hedges, and rhetorical flourishes), whereas core derivational tokens (e.g., operators, equations, intermediate results) retain high confidence. This indicates that MoT is essentially **weakening the learning of inductive bias** of different teachers, while repeatedly **reinforcing the learning of consensus reasoning ability**. We also detail token-level confidence for MoT and STD(R1) on R1-distilled CoTs in the Appendix Figs. 7 and Figs. 8.

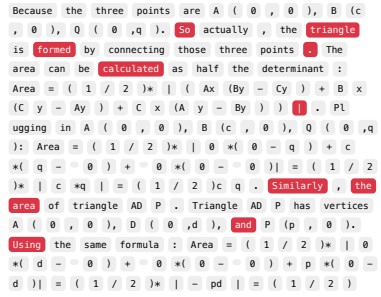

Figure 5: Tokens marked with confidence drops relative to the Base model after MoT.

**MoT mitigates inter-teacher conflicts and trains in a flatter loss landscape.** We design two complementary evaluations with clear goals: (i) a theoretical "two-bonus" decomposition to test whether expert-wise preconditioning boosts the useful driving term and whether cross-teacher interference is provably reduced—thereby explaining conflict mitigation and flatter updates than MTD; and (ii) a linear mode connectivity probe to check loss landscape flatness of MoT. The "two-bonus" decomposition (Appendix A) shows that expert-wise preconditioning of MoT boosts the driving term while a contractive bound reduces cross-teacher interference, with averaging further shrinking the curvature penalty. Complementarily, a linear mode connectivity probe (Appendix D) indicates that MoT yields markedly smoother loss curves than MTD, indicating flatter regions and reduced sensitivity to teacher noises.

# 9 CONCLUSION

We presented **Merge-of-Thought Distillation** (MoT), a lightweight framework that unifies supervision from multiple heterogeneous teachers for long chain-of-thought (CoT) reasoning by alternating teacher-specific SFT with weight-space merging. Revisiting teacher selection shows that different students have different "best teachers," and even the same student's best teacher varies across datasets; MoT sidesteps brittle manual selection by fusing complementary reasoning abilities into a single student. With only about **200** CoT samples, applying MoT to a Qwen3-14B student surpasses Deepseek-R1, Qwen3-32B, and OpenAI-O1. Besides, MoT consistently beats the best single-teacher and naive multi-teacher unions, improves general reasoning while mitigating catastrophic forgetting, and is robust to distribution-shifted and peer-level teachers. Finally, we provide theoretical and empirical evidence that MoT naturally induces a consensus CoT by eliminating teacher-specific inductive biases and inter-teacher conflicts while repeatedly reinforcing the learning of consensus reasoning feature, which enables training in a flatter region of the loss landscape and effective transfer to new student models.

**Ethics Statement.** We affirm compliance with the ICLR Code of Ethics. Our study does not involve human subjects or personally identifiable information. Training/evaluation use public math/QA benchmarks (e.g., AIME24/25, CEVAL, MMLU variants, GPQA, LiveCodeBench, Phy-Bench) under their respective licenses; we follow all license terms and cite original sources. Teacher trajectories (CoTs) are generated by publicly available LLMs and filtered to remove potential toxicity. No sensitive domains (medical/financial/legal advice) are targeted. We report all compute details to support efficient replication. Any conflicts of interest or sponsorship will be disclosed per ICLR policy at camera-ready; none are known that would bias the results at submission time.

**Reproducibility Statement.** We take reproducibility seriously. The method is fully specified in Section 4, with training schedules and hyperparameters in Appendix G.1 and ablations in Table 4. We average AIME over 16 seeds and save checkpoints every 50 steps; full per-step results are reported in Appendix G. To facilitate exact reruns, we release (anonymized) artifacts as supplementary material: main code, training scripts, data preprocessing steps, and environment requirement files (conda). Appendix 5 details dataset sources, splits, and filtering; Appendix G.1 lists hardware. These materials allow independent reproduction of tables and figures without additional calibration.

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

## A  THEORETICAL ANALYSIS

In this section, we will provide a detailed theoretical analysis to explain the advantages of MoT over MTD in addressing conflicts and mitigating forgetting issues.

Our analysis is based on a comparison of **the gradient update processes of MoT and MTD**.

**Preliminary.**  We approximate the model update for each expert by using second-order Taylor expansion:

$$\ell_k(\theta) \approx \ell_k(\theta_{t-1}) + g_k^\top (\theta - \theta_{t-1}) + \tfrac{1}{2} (\theta - \theta_{t-1})^\top H_k(\theta - \theta_{t-1}),$$

where $g_k = \nabla \ell_k(\theta_{t-1})$, $H_k = \nabla^2 \ell_k(\theta_{t-1})$, and $\ell_k(\theta)$ is the loss function for expert $k$ evaluated at point $\theta$. We also define the mixture gradient and Hessian as weighted sums of the individual gradients and Hessians:

$$\bar{g} = \sum_k \alpha_k g_k, \quad \bar{H} = \sum_k \alpha_k H_k,$$

where $\alpha_k \geq 0$ and $\sum_k \alpha_k = 1$ are the weights assigned to each expert.

Each branch performs $E_k$ steps of gradient descent with a stepsize $\eta$ starting from $\theta_{t-1}$. Based on second-order Taylor expansion, we have $\theta_{k,E} = \theta_{t-1} - P_k g_k$, $P_k = \eta \sum_{e=0}^{E-1} (I - \eta H_k)^e$, where $P_k$ is the "preconditioner" used in each branch's local optimization process.

We also have the below closed-form solution for preconditioner:

$$P_k = s_{E_k}(H_k), \qquad s_E(\lambda) = \frac{1 - (1 - \eta\lambda)^E}{\lambda} = \eta \sum_{e=0}^{E-1} (1 - \eta\lambda)^e,$$

where $s_E(\lambda)$ represents the effective step size along the direction defined by the eigenvalue $\lambda$ of the Hessian matrix $H_k$.

The expression for $s_E(\lambda)$ can be derived by considering the update rule for gradient descent in the presence of a Hessian, where each step of gradient descent applies a scaling factor depending on the eigenvalue $\lambda$ of the Hessian matrix at each iteration. For large $E$ or small $\eta\lambda$, $s_E(\lambda)$ approximates the inverse of the eigenvalue $\lambda$, leading to more efficient updates along lower-curvature directions.

Hence, the branch displacement for expert $k$ is given by:

$$\delta_k = -P_k g_k = -s_{E_k}(H_k) g_k,$$

and the MoT merge, which aggregates the displacements from all experts, is:

$$\Delta = \sum_k \alpha_k \delta_k = - \sum_k \alpha_k P_k g_k.$$

For the MTD, which also runs $E$ local steps at the same anchor point, the preconditioner is defined as:

$$P_{\mathrm{mtd}} = s_E(\bar{H}),$$

where $\bar{H}$ is the weighted sum of the Hessians of all experts, and the $E$-step update is:

$$- P_{\mathrm{mtd}}\bar{g} = -s_E(\bar{H})\bar{g}.$$

Here, $P_{\mathrm{mtd}}$ is the preconditioner used for the mixture of experts, and $\bar{g}$ is the mixture gradient.

**Assumption 1** (Local quadratic & stable steps). *Each $\ell_k$ is $C^2$ in a neighborhood $\mathcal{N}$ of $\theta_{t-1}$. Let $H_k = \nabla^2 \ell_k(\theta_{t-1})$ and $L_{\max} = \max_k \lambda_{\max}(H_k)$. We choose a stepsize $\eta \in (0, 2/L_{\max})$ and run $E_k \geq 1$ local steps whose iterates remain in $\mathcal{N}$.*

**Two bonuses on the linear part.**  The one-round improvement under the quadratic surrogate $F_Q(\delta) = \bar{g}^\top \delta + \frac{1}{2}\delta^\top \bar{H}\delta$ splits into a *linear* "driving" term and a *quadratic* penalty. For the linear

term we have the following variance-type decompositions:

$$\underbrace{\Big\|\sum_k \alpha_k P_k g_k\Big\|^2}_{\text{MoT linear}} = \underbrace{\sum_k \alpha_k \langle g_k, P_k g_k\rangle}_{\text{expert-wise preconditioning}} - \underbrace{\tfrac{1}{2}\sum_{i,j}\alpha_i\alpha_j\|P_i g_i - P_j g_j\|^2}_{I_{\text{mot}}\ \geq 0}, \tag{3}$$

$$\underbrace{\|\bar g\|^2_{P_{\text{mtd}}}}_{\text{mtd linear}} = \underbrace{\sum_k \alpha_k \langle g_k, P_{\text{mtd}} g_k\rangle}_{\text{single preconditioner}} - \underbrace{\sum_k \alpha_k \|g_k - \bar g\|^2_{P_{\text{mtd}}}}_{I_{\text{mtd}}(P_{\text{mtd}})\ \geq 0}, \tag{4}$$

where $\|x\|^2_M = x^\top M x$. Subtracting equation 4 from equation 3 yields the *two-bonus* difference

$$\Delta_{\text{lin}} := \Big\|\sum_k \alpha_k P_k g_k\Big\|^2 - \|\bar g\|^2_{P_{\text{mtd}}}$$

$$= \underbrace{\sum_k \alpha_k \langle g_k, (P_k - P_{\text{mtd}})g_k\rangle}_{\text{(A') preconditioning gain}} + \underbrace{I_{\text{mtd}}(P_{\text{mtd}}) - I_{\text{mot}}}_{\text{(B') interference mitigation}}. \tag{5}$$

**When is (A') $\geq 0$?**

**Lemma 1** (Monotonicity of $s_E$). *For any fixed $E \geq 1$ and $\eta > 0$, $s_E(\lambda) = \eta\sum_{e=0}^{E-1}(1-\eta\lambda)^e$ is strictly decreasing in $\lambda$ on $(0, 2/\eta)$.*

If $H_k$ and $\bar H$ are (approximately) simultaneously diagonalizable, then $D_k := \langle g_k, (P_k - P_{\text{mtd}})g_k\rangle = \|g_k\|^2 \sum_r w_{k,r}\big(s_E(\lambda_{k,r}) - s_E(\bar\lambda_r)\big)$, with weights $w_{k,r} = \frac{(q_r^\top g_k)^2}{\|g_k\|^2}$. Hence $D_k \geq 0$ whenever most weight lies on directions where $\lambda_{k,r} \leq \bar\lambda_r$. Aggregating with $\alpha_k$ gives (A')$\geq 0$.

**When is (B') $\geq 0$?  A contractive bound on interference.** Let $\mathcal{S} = \text{span}\{g_i - g_j\}_{i,j}$ be the disagreement subspace.

**Assumption (direction-wise contraction on $\mathcal{S}$).** There exists $\rho \in (0,1]$ such that on $\mathcal{S}$ one of the following equivalent conditions holds:

*(Coord.)* $H_k$ and $\bar H$ are (approximately) simultaneously diagonalizable on $\mathcal{S}$ with eigenbasis $\{q_r\}$; let $p_{k,r} = s_E(\lambda_{k,r})$ and $p_{\text{mtd},r} = s_E(\bar\lambda_r)$. For all $r$ with $q_r \in \mathcal{S}$,
$$\max_i p_{i,r} \leq \rho\, p_{\text{mtd},r}.$$

*(Basis-free)* For all $v \in \mathcal{S}$ and all $k$,
$$\|P_k v\|^2 \leq \rho^2\|v\|^2_{P_{\text{mtd}}} \qquad (\text{i.e., } v^\top P_k^\top P_k v \leq \rho^2\, v^\top P_{\text{mtd}} v).$$

The above is natural on high-curvature/disagreement directions because $s_E(\lambda)$ is decreasing in $\lambda$: along directions where at least one expert has directional curvature no smaller than the mixture (a common empirical pattern), its preconditioning coefficient is smaller, yielding stronger contraction.

Under this assumption we have
$$I_{\text{mot}} = \tfrac{1}{2}\sum_{i,j}\alpha_i\alpha_j\|P_i g_i - P_j g_j\|^2 \leq \rho^2\,\tfrac{1}{2}\sum_{i,j}\alpha_i\alpha_j\|g_i - g_j\|^2_{P_{\text{mtd}}} = \rho^2\, I_{\text{mtd}}(P_{\text{mtd}}). \tag{6}$$

Hence (B') $= I_{\text{mtd}}(P_{\text{mtd}}) - I_{\text{mot}} \geq (1-\rho^2)\,I_{\text{mtd}}(P_{\text{mtd}}) \geq 0$.

**Implicit shrinkage from averaging enters the quadratic penalty.** With $\Delta = -\sum_k \alpha_k P_k g_k$, the quadratic penalties satisfy

$$R_{\text{mot}} = \tfrac{1}{2}\Delta^\top \bar H \Delta \leq \tfrac{1}{2}\lambda_{\max}(\bar H)\Big(\sum_k \alpha_k\|P_k g_k\|^2 \underbrace{- I_{\text{mot}}}_{\text{shrinkage from averaging}}\Big), \tag{7}$$

$$R_{\text{mtd}} = \tfrac{1}{2}\eta^2\,\bar g^\top \bar H\,\bar g \leq \tfrac{1}{2}\lambda_{\max}(\bar H)\Big(\sum_k \alpha_k\|P_{\text{mtd}} g_k\|^2 - I_{\text{mtd}}(P_{\text{mtd}})\Big). \tag{8}$$

Note the *minus* interference terms, showing that averaging contracts the update norm and directly reduces the curvature penalty.

Table 9: MoT applied to Llama-3.1-8B-Instruct on Math500 under the BOBA-200 setting. MoT consistently outperforms all single-teacher STD variants, indicating that MoT is not tied to Qwen-specific design choices.

| Method | Math500 score |
|---|---|
| Base | 49.65 |
| STD (Qwen3-32B) | 55.45 |
| STD (QWQ) | 55.80 |
| STD (Qwen3-235B) | 57.45 |
| STD (Deepseek-R1) | 53.80 |
| **MoT (ours)** | **62.65** |

**Net one-round advantage.** Combining equation 5–equation 6 and the penalty bounds yields

$$\underbrace{\Delta_{\text{mot}} - \Delta_{\text{mtd}}}_{\text{MoT minus MTD}} \gtrsim \underbrace{\sum_k \alpha_k \langle g_k, (P_k - P_{\text{mtd}}) g_k \rangle}_{\text{(A')}} + \underbrace{(1 - \rho^2) I_{\text{mtd}}(P_{\text{mtd}})}_{\text{(B')}} - \frac{1}{2} \lambda_{\max}(\bar{H}) \cdot \Big[ \cdots \Big],$$

where $[\cdots]$ gathers the (usually small in the stable regime) difference of squared update norms. Thus, under gradient/curvature heterogeneity and stable steps, MoT enjoys a larger linear driving term (A') and smaller interference (B'), while averaging further cuts the quadratic penalty.

**Special case $E = 1$ (for reference).** Then $P_k = P_{\text{mtd}} = \eta I$, and equation 5 reduces to the familiar two-term decomposition

$$\underbrace{\sum_k \alpha_k f_k \|g_k\|^2 - \eta \|\bar{g}\|^2}_{f_k = \eta} = \underbrace{0}_{\text{(A)}} + \eta \underbrace{\sum_k \alpha_k \|g_k - \bar{g}\|^2}_{\text{(B) variance bonus}}.$$

**Remark 1** (Implicit proximal effect (Mitigating Forgetting)). *The matrix series identity $P_k = \eta \sum_{e=0}^{E_k - 1} (I - \eta H_k)^e$ shows a direction-dependent shrink toward the anchor; in each eigendirection $\lambda$ the effective step is $s_E(\lambda)$, **larger for low curvature and smaller for high curvature**, explaining MoT's stability without explicit proximal terms.*

**Remark 2** (Unified Improvements (Mitigating Conflicts)). *A positive value for both bonus terms indicates that MoT reduces gradient interference and produces a larger effective update, thereby improving optimization progress.*

A.1 ADDITIONAL ANALYSES: GENERALIZATION AND ROBUSTNESS OF MOT

**Generalization to other backbones.** To examine whether MoT is specific to the Qwen family or can transfer to other architectures, we replicate the BOBA-200 setup on a different backbone, **Llama-3.1-8B-Instruct**. We use exactly the same teacher pool, data, and MoT procedure, and evaluate on Math500. The results are shown in Table 9.

MoT significantly improves the Llama-3.1-8B-Instruct backbone and provides a sizable margin over the *best* single-teacher distillation, supporting the view that MoT is a lightweight, architecture-agnostic training procedure rather than a Qwen-specific trick.

**Seed sensitivity and early stopping.** We further study the robustness of MoT to random seeds and early-stopping choices. On Qwen3-8B with BOBA-200, we run MoT with 5 independent seeds under the same 5-round schedule, and report the AIME average (AIME AVG = (AIME24 + AIME25)/2) at each round. In addition, we implement a fixed validation-based early-stopping rule for MoT: 10% of the original training set is held out as a validation set, and for each run we select the checkpoint (across rounds) with the best validation score and then report its test performance. The results are summarized in Table 10.

The peak around merge round 4 is stable across seeds, without any "best-of" checkpoint selection. Validation-based early stopping yields slightly lower AIME AVG than always using round 4 (as expected, since the effective training set is smaller), but remains strong and better than any STD/MTD

Table 10: MoT robustness across seeds and rounds on Qwen3-8B + BOBA-200. We report AIME AVG (AIME24/25 average) as mean $\pm$ std over 5 seeds. "Early-stopping" denotes validation-based selection using a 10% held-out split.

| Configuration | AIME AVG (mean $\pm$ std) |
|---|---|
| Merge round 1 | $70.17 \pm 0.63$ |
| Merge round 2 | $72.37 \pm 0.65$ |
| Merge round 3 | $73.38 \pm 0.09$ |
| Merge round 4 | $\mathbf{74.89 \pm 1.05}$ |
| Merge round 5 | $72.75 \pm 0.81$ |
| Early-stopping (val-based) | $73.46 \pm 0.31$ |

Table 11: MoT on a code reasoning domain (LiveCodeBench) using 178 code-domain examples. MoT again outperforms the best single-teacher STD.

| Method | LiveCodeBench score |
|---|---|
| Base | 55.76 |
| STD (32B) | 58.08 |
| STD (QWQ) | 56.88 |
| STD (235B) | 58.89 |
| STD (R1) | 53.89 |
| **MoT (ours)** | **61.08** |

baseline, and it also improves over taking the same final checkpoint of MoT without early-stopping. Overall, these results indicate that MoT's gains are robust to random seeds and remain effective under a fixed, validation-based early-stopping rule.

**Generalization beyond mathematical reasoning.** To evaluate whether MoT extends beyond competition math, we consider a code reasoning domain using 178 code-domain examples from Deng et al. (2025) and apply exactly the same distillation, training, and evaluation pipeline as in the main experiments (same teacher pool, same 1Q–multiA CoT collection, same MoT procedure). We evaluate on LiveCodeBench, and report the results in Table 11.

As in the mathematical reasoning setting, MoT again outperforms the best single-teacher STD, suggesting that MoT is not restricted to math and can also improve code reasoning under the same multi-teacher long-CoT setup. Furthermore, as reported in Section 6.3, MoT improves performance on a range of general benchmarks (e.g., CEVAL, MMLU variants, physics and coding benchmarks) while incurring smaller drops on catastrophic-forgetting–sensitive tasks compared to the best single-teacher STD. Together, these results provide concrete evidence that MoT generalizes beyond competition math to other domains and evaluation suites.

**Additional math benchmark: HMMT.** To further diversify mathematical evaluation, we also evaluate Qwen3-8B distilled on BOBA-200 using MoT and four single-teacher STDs on the HMMT benchmark. Results are shown in Table 12.

Here, QWQ remains the best single teacher, consistent with Table 1 for this student/dataset configuration, which supports the stability of our teacher-selection analysis under a fixed setting. Importantly, MoT still achieves the highest score, improving over the best STD and reinforcing that MoT effectively unifies multiple teachers' reasoning abilities and raises the student's reasoning ceiling.

**Summary across math and code domains.** Across different evaluation domains—AIME24/25, HMMT, and the code-reasoning setting—we observe that the identity of the "best" teacher changes with the dataset or domain (e.g., QWQ vs. Qwen3-235B), supporting our claim that teacher choices are not universal. At the same time, MoT consistently outperforms all single-teacher STDs in these settings, confirming the effectiveness of our multi-teacher consensus distillation. A broader sweep

Table 12: Performance of Qwen3-8B on HMMT after distillation on BOBA-200. QWQ remains the strongest single teacher, whereas MoT achieves the best overall score.

| Method | HMMT score |
|---|---|
| Base | 38.33 |
| STD (32B) | 43.33 |
| STD (QWQ) | **48.33** |
| STD (235B) | 45.83 |
| STD (R1) | 40.83 |
| **MoT (ours)** | **52.50** |

Table 13: Advanced merging baselines and MoT variants on Qwen3-8B + BOBA-200. TIES and DARE are used both as one-shot merges and as merge operators inside MoT.

| Method | AIME AVG |
|---|---|
| Base | 71.46 |
| One-shot TIES | 71.67 |
| One-shot DARE | 60.42 |
| **MoT (TIES)** | 73.34 |
| **MoT (DARE)** | 74.17 |
| **MoT (simple merge, ours)** | **74.48** |

over additional domains (e.g., large-scale scientific QA) is left for future work, but the new code and HMMT experiments already provide further evidence beyond the original math benchmarks.

## A.2 EFFECT OF MERGING OPERATOR AND NUMBER OF TEACHERS

**Advanced merging operators: TIES and DARE.** To compare MoT against more advanced model-merging and data-fusion techniques, we incorporate several recent operators into our pipeline. On Qwen3-8B + BOBA-200, we evaluate: (i) one-shot TIES merging, (ii) one-shot DARE merging, and (iii) MoT variants that replace simple averaging with TIES or DARE in the merge step. All other settings (teachers, data, schedule, evaluation) are kept identical. The final AIME average (AIME24/25) is reported in Table 13. We also report the per-round behavior of MoT(TIES) and MoT(DARE) in Tables 14 and 15.

These results lead to three observations:
(1) First, advanced one-shot merges alone are not sufficient in our setting: one-shot TIES brings only a minor gain over the base model, and one-shot DARE causes a severe performance drop. This suggests that techniques designed for merging models trained on different domains or tasks may be much less directly suitable for unifying different reasoning paths for the same questions.
(2) Second, when TIES or DARE is used inside the MoT loop, performance improves substantially, with DARE gaining almost +14 points over its one-shot counterpart. Algorithmically, this is consistent with the fact that one-shot DARE acts on highly conflicting teacher-specific updates all at once (leading to over-pruning of partially misaligned but useful directions), whereas DARE inside MoT sees smaller, progressively more aligned deltas across rounds and branches, and thus behaves like a gradual consensus regularizer that keeps directions repeatedly reinforced by multiple teachers.
(3) Third, simple averaging still achieves the highest and most robust ceiling: MoT(TIES) and MoT(DARE) tend to converge faster across rounds but plateau at a slightly lower level or overfit more, while MoT with plain averaging attains the best final AIME AVG. A plausible explanation, consistent with our analysis in Section 7, is that simple averaging does not impose any parameter filtering, allowing MoT to naturally absorb useful signals even from suboptimal or noisy teachers and thereby achieve a higher reasoning ceiling.

**Effect of the number of teachers.** We also study how MoT behaves as we vary the number of teachers. On BOBA-200 with Qwen3-8B, we start from the best single teacher (by STD perfor-

Table 14: MoT(TIES) across merge rounds on Qwen3-8B + BOBA-200.

| Round | AIME24 | AIME25 | AIME AVG |
|-------|--------|--------|----------|
| 1 | 73.33 | 66.67 | 70.00 |
| 2 | 77.50 | 67.50 | 72.50 |
| 3 | 77.50 | 69.17 | 73.34 |
| 4 | 74.17 | 65.00 | 69.59 |
| 5 | 75.00 | 65.83 | 70.42 |

Table 15: MoT(DARE) across merge rounds on Qwen3-8B + BOBA-200.

| Round | AIME24 | AIME25 | AIME AVG |
|-------|--------|--------|----------|
| 1 | 75.00 | 62.50 | 68.75 |
| 2 | 76.67 | 65.00 | 70.84 |
| 3 | 78.33 | 70.00 | 74.17 |
| 4 | 79.17 | 69.17 | 74.17 |
| 5 | 73.33 | 65.83 | 69.58 |

mance), then progressively add the second-best, third-best, and finally the noisy teacher R1 into the MoT pool, keeping the MoT configuration fixed. The results are shown in Table 16.

We observe a monotonic improvement in AIME AVG as more teachers are added, and performance continues to increase even after including the distribution-shifted/noisy teacher R1. This supports the view that, in our experimental regime, MoT can effectively extract complementary signals from additional teachers and is robust enough to benefit from them. We do not claim that this behavior will persist for arbitrarily large pools of low-quality or adversarial teachers; in such extreme cases, stronger filtering or adaptive weighting would likely be necessary. Systematically studying how performance scales with larger and more heterogeneous teacher sets is an interesting direction for future work.

## B  LIMITATIONS

(1) We currently merge branches via simple uniform parameter averaging; future work will explore alternative merge strategies.

(2) Beyond AIME24/25, there is a lack of sufficiently challenging math benchmarks, which limits evaluation depth on high-difficulty mathematical reasoning.

(3) Baseline results in the main results are taken from the original papers/reports because many baselines do not release code/models or disclose key training details like data curation or key hyper-parameters. Consequently, they were not re-evaluated under a unified, consistent evaluation configuration, which may affect strict comparability.

## C  BETTER STUDENT IS A BETTER TEACHER

Table 17: Student-as-teacher distillation on BOBA-200. Teachers are base model or student models obtained with Best-STD/MoT. We report raw scores on reasoning benchmarks mentioned earlier.

| Teacher model | Student model | Teacher Config | AIME24 | AIME25 | PhyBench | LiveCodeBench | GPQA-Diamond | AVG |
|---------------|---------------|----------------|--------|--------|----------|---------------|--------------|-----|
| Qwen3-14B | Qwen3-8B | Base | 74.17 | 67.08 | 23.06 | **58.98** | 57.80 | 56.22 |
| | | Best STD | 75.21 | 64.17 | 23.74 | 56.74 | 58.33 | 55.64 |
| | | MoT | **75.63** | **68.96** | **24.28** | 58.83 | **59.22** | **57.38** |
| Qwen3-30B-A3B | Qwen3-14B | Base (Vanilla) | 79.17 | 68.96 | 28.31 | 61.41 | 61.65 | 59.90 |
| | | Best STD | 77.08 | **71.88** | 29.40 | **63.36** | 61.87 | 60.72 |
| | | MoT | **80.00** | 71.67 | **29.63** | 62.99 | **62.69** | **61.40** |

Table 16: Effect of the number of teachers in MoT on Qwen3-8B + BOBA-200. Teachers are added in descending order of single-teacher STD performance, with R1 being the most distribution-shifted/noisy teacher.

| # Teachers in MoT | AIME24 | AIME25 | AIME AVG |
|---|---|---|---|
| 1 (best teacher) | 76.25 | 67.50 | 71.88 |
| 2 | 77.50 | 68.33 | 72.92 |
| 3 | 78.13 | 69.17 | 73.65 |
| 4 (with R1) | 78.33 | 70.63 | 74.48 |

## D  PROBING LOSS-LANDSCAPE FLATNESS VIA BASE-TO-CHECKPOINT INTERPOLATION: MoT VS. MTD

**Setup and purpose.** To assess how stably a trained model sits in parameter space, we probe **loss-landscape flatness** via linear mode connectivity (LMC) between the *base* model and the final trained checkpoint (from either MTD or our MoT). For $\lambda \in [0, 1]$, we define

$$\theta(\lambda) \ = \ \lambda\,\theta_{\text{base}} \ + \ (1 - \lambda)\,\theta_{\text{ckpt}},$$

so that $\lambda=1$ recovers the base model and $\lambda=0$ recovers the trained checkpoint. At each $\lambda$ on a fixed grid, we evaluate AIME24 (pass@1, 64-run average). A smooth/high trajectory indicates a flatter, more robust region with fewer barriers; a sharp/erratic trajectory suggests a bumpier landscape and stronger interference among supervision signals.

**Findings.** On both BOBA-200 and S1K-200 with the 8B student, MoT yields a **much smoother** and more stable performance curve than MTD as $\lambda$ varies: performance rises steadily toward the checkpoint and decays gradually away from it. This behavior is consistent with MoT training in a **flatter** region (greater robustness to weight perturbations) and **better reconciliation of cross-teacher supervision conflicts**. In contrast, MTD exhibits steeper drops and local irregularities, implying residual inter-teacher interference.

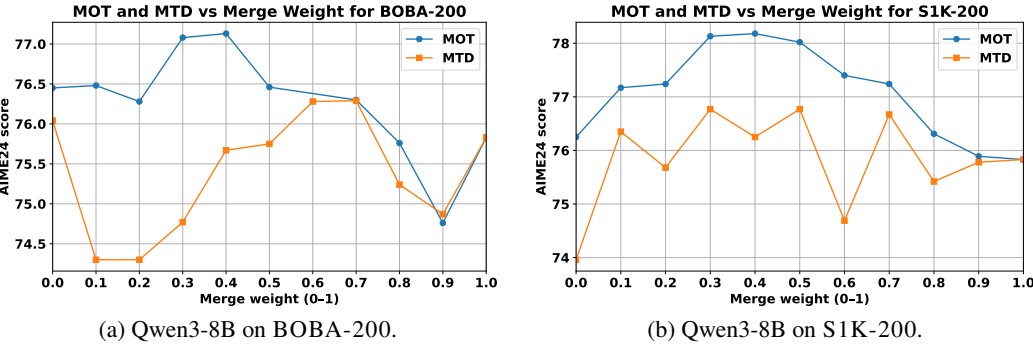

(a) Qwen3-8B on BOBA-200.  (b) Qwen3-8B on S1K-200.

Figure 6: Base-to-checkpoint linear interpolation (LMC). MoT shows smoother, higher trajectories than MTD on AIME24, indicating a flatter loss region and more robust training.

## E  TASK-TYPE BREAKDOWN ACROSS STD/MTD/MoT

**Setup and goal.** We provide a consolidated evaluation on **BOBA-200** across *all* STD/MTD settings alongside MoT, covering nine benchmarks: catastrophic-forgetting–sensitive (CEV/SG/IFE), reasoning-knowledge (SQ/MP/MR), and pure reasoning (PB/LCB/GPQA-D). For each setting, we report raw scores and *group-wise average changes* versus the same-scale Base: "Avg drop (cat.)" for catastrophic-forgetting–sensitive tasks (negative indicates a drop), "Avg gain (reason.)" for reasoning-knowledge tasks, and "Avg gain (pure)" for pure reasoning tasks. We observe a trade-off among STD choices (stronger reasoning vs. better forgetting mitigation), while **MoT** simultaneously yields strong math/general reasoning gains and *significantly* mitigates catastrophic forgetting.

**Summary.** Results are shown in Table 18, Table 19 and Table 20. Single-teacher choices present a clear trade-off: some teachers maximize reasoning gains but induce larger average drops on forgetting-sensitive tasks, while others better preserve foundational abilities but yield smaller reasoning gains. **MoT** alleviates this tension: it delivers strong improvements on reasoning-knowledge and pure reasoning benchmarks, while reducing average drops on forgetting-sensitive tasks across scales.

Table 18: Catastrophic-forgetting–sensitive tasks on BOBA-200 (CEV / SG / IFE). "Avg drop (cat.)" is the average change vs. the same-scale Base (negative indicates a drop). For Qwen3-30B-A3B, SG for *STD(QWQ)* is unavailable ("—"); the average uses available metrics (CEV & IFE) and compares to Base on the same subset.

| Base model | Setting | CEV | SG | IFE | Avg drop (cat.) |
|---|---|---|---|---|---|
| **Qwen3-8B** | Base | 83.58 | 10.51 | 83.60 | – |
| | STD (Qwen3-32B) | 81.35 | 10.38 | 81.34 | ↓1.54 |
| | STD (Qwen3-235B) | 83.28 | 9.57 | 81.18 | ↓1.22 |
| | STD (QWQ) | 83.43 | 9.97 | 81.62 | ↓0.89 |
| | STD (Deepseek-R1) | 83.06 | 9.70 | 81.79 | ↓1.05 |
| | MTD (All Teachers) | 83.14 | 10.04 | 82.07 | ↓0.81 |
| | MoT (ours) | 83.73 | 10.15 | 82.04 | ↓0.59 |
| **Qwen3-14B** | Base | 86.78 | 10.76 | 84.69 | – |
| | STD (Qwen3-32B) | 84.55 | 10.07 | 82.76 | ↓1.62 |
| | STD (Qwen3-235B) | 83.73 | 10.26 | 82.56 | ↓1.89 |
| | STD (QWQ) | 83.73 | 10.22 | 82.36 | ↓1.97 |
| | STD (Deepseek-R1) | 84.32 | 9.92 | 82.91 | ↓1.69 |
| | MTD (All Teachers) | 85.14 | 9.88 | 82.22 | ↓1.66 |
| | MoT (ours) | 86.70 | 10.38 | 83.51 | ↓0.55 |
| **Qwen3-30B-A3B** | Base | 85.88 | 10.66 | 83.76 | – |
| | STD (Qwen3-32B) | 85.74 | 9.93 | 82.32 | ↓0.77 |
| | STD (Qwen3-235B) | 84.18 | 10.02 | 80.44 | ↓1.89 |
| | STD (QWQ) | 83.80 | 9.65 | 80.03 | ↓2.27 |
| | STD (Deepseek-R1) | 83.36 | 9.31 | 80.61 | ↓2.34 |
| | MTD (All Teachers) | 84.55 | 10.12 | 79.77 | ↓1.95 |
| | MoT (ours) | 86.55 | 10.52 | 83.54 | ↑0.10 |

Table 19: Reasoning-related tasks on BOBA-200 (SQ / MP / MR). "Avg gain (reason.)" is the average change vs. the same-scale Base (positive indicates an increase).

| Base model | Setting | SQ | MP | MR | Avg gain (reason.) |
|---|---|---|---|---|---|
| **Qwen3-8B** | Base | 32.31 | 71.42 | 83.21 | – |
| | STD (Qwen3-32B) | 34.37 | 73.05 | 84.82 | ↑1.77 |
| | STD (Qwen3-235B) | 32.63 | 72.83 | 84.84 | ↑1.12 |
| | STD (QWQ) | 33.88 | 72.00 | 84.42 | ↑1.12 |
| | STD (Deepseek-R1) | 33.88 | 70.92 | 84.21 | ↑0.69 |
| | MTD (All Teachers) | 33.60 | 72.34 | 84.65 | ↑1.22 |
| | MoT (ours) | 34.44 | 73.30 | 84.42 | ↑1.74 |
| **Qwen3-14B** | Base | 32.61 | 75.26 | 85.74 | – |
| | STD (Qwen3-32B) | 32.31 | 75.36 | 85.93 | ↓0.00 |
| | STD (Qwen3-235B) | 32.17 | 74.71 | 86.37 | ↓0.12 |
| | STD (QWQ) | 32.42 | 74.76 | 85.19 | ↓0.41 |
| | STD (Deepseek-R1) | 32.63 | 74.04 | 86.04 | ↓0.30 |
| | MTD (All Teachers) | 32.77 | 74.97 | 85.82 | ↓0.02 |
| | MoT (ours) | 32.65 | 75.59 | 86.53 | ↑0.39 |
| **Qwen3-30B-A3B** | Base | 31.68 | 75.26 | 85.81 | – |
| | STD (Qwen3-32B) | 32.26 | 76.12 | 86.46 | ↑0.70 |
| | STD (Qwen3-235B) | 31.52 | 75.96 | 86.04 | ↑0.26 |
| | STD (QWQ) | 32.24 | 75.28 | 84.86 | ↓0.12 |
| | STD (Deepseek-R1) | 33.00 | 72.55 | 84.16 | ↓1.01 |
| | MTD (All Teachers) | 32.31 | 74.75 | 86.67 | ↑0.33 |
| | MoT (ours) | 32.26 | 76.21 | 86.74 | ↑0.82 |

Table 20: Pure reasoning tasks on BOBA-200 (PB / LCB / GPQA-D). "Avg gain (pure)" is the average change vs. the same-scale Base (positive indicates an increase).

| Base model | Setting | PB | LCB | GPQA-D | Avg gain (pure) |
|---|---|---|---|---|---|
| **Qwen3-8B** | Base | 20.47 | 55.76 | 57.77 | – |
| | STD (Qwen3-32B) | 23.19 | 59.06 | 57.42 | ↑1.89 |
| | STD (Qwen3-235B) | 23.17 | 57.90 | 58.11 | ↑1.73 |
| | STD (QWQ) | 22.85 | 59.88 | 59.85 | ↑2.86 |
| | STD (Deepseek-R1) | 21.90 | 56.78 | 56.50 | ↑0.39 |
| | MTD (All Teachers) | 22.47 | 54.79 | 60.32 | ↑1.19 |
| | MoT (ours) | 24.07 | 58.79 | 60.54 | ↑3.13 |
| **Qwen3-14B** | Base | 28.53 | 61.41 | 60.83 | – |
| | STD (Qwen3-32B) | 30.72 | 62.84 | 61.52 | ↑1.44 |
| | STD (Qwen3-235B) | 30.61 | 63.21 | 63.79 | ↑2.28 |
| | STD (QWQ) | 28.36 | 62.80 | 63.44 | ↑1.28 |
| | STD (Deepseek-R1) | 27.29 | 61.15 | 62.91 | ↑0.19 |
| | MTD (All Teachers) | 29.51 | 58.50 | 63.19 | ↑0.14 |
| | MoT (ours) | 30.77 | 63.59 | 64.26 | ↑2.62 |
| **Qwen3-30B-A3B** | Base | 28.57 | 61.08 | 59.76 | – |
| | STD (Qwen3-32B) | 33.43 | 61.79 | 60.48 | ↑2.10 |
| | STD (Qwen3-235B) | 33.31 | 61.34 | 61.81 | ↑2.35 |
| | STD (QWQ) | 32.44 | 60.74 | 60.32 | ↑1.36 |
| | STD (Deepseek-R1) | 29.31 | 59.02 | 59.66 | ↓0.47 |
| | MTD (All Teachers) | 32.50 | 56.85 | 61.33 | ↑0.42 |
| | MoT (ours) | 33.46 | 62.54 | 62.34 | ↑2.98 |

# F DATASET

Table 21: STD and MTD distillation datasets derived from BOBA-200 and S1K-200.

| Source | Teacher | Distillation dataset name | Size |
|---|---|---|---|
| BOBA-200 | QWQ | BOBA-200-QWQ | 195 |
| | Qwen3-32B | BOBA-200-32B | 191 |
| | Qwen3-235B | BOBA-200-235B | 197 |
| | Deepseek-R1 | BOBA-200-R1 | 198 |
| | ALL TEACHERS | BOBA-200-MTD | 781 |
| S1K-200 | QWQ | S1K-200-QWQ | 161 |
| | Qwen3-32B | S1K-200-32B | 164 |
| | Qwen3-235B | S1K-200-235B | 169 |
| | Deepseek-R1 | S1K-200-R1 | 168 |
| | ALL TEACHERS | S1K-200-MTD | 662 |

# G ADDITIONAL TRAINING DETAILS AND FULL ABLATIONS

## G.1 TRAINING HYPERPARAMETERS

Unless otherwise noted, all experiments follow a shared set of training choices designed for long chain-of-thought (CoT) sequences and stable optimization:

- Model/input formatting: We use the Qwen3 instruction template to format prompts and responses consistently across datasets.

- Context length: The maximum sequence length is 25k tokens to accommodate long CoT traces with minimal truncation.

- Precision and kernels: Training uses bfloat16 with FlashAttention-2 to improve memory efficiency and throughput for long contexts.

- Optimizer and schedule: AdamW with betas (0.9, 0.95), weight decay 0.1, cosine learning-rate schedule with a base learning rate of 1e-5 and 1% warmup. Gradients are clipped at a norm of 1.0 for stability.

- Batch and accumulation: We train on $8\times$ H800 GPUs with a per-device batch size of 1 and gradient accumulation of 8, resulting in an effective batch size of 64 sequences per optimization step.

- Logging and checkpointing: We log every step and save a checkpoint every 50 steps; up to 10 most recent checkpoints are kept, and only model weights are saved to reduce I/O overhead.

Protocol-specific details:

- MoT: One "round" consists of 50 optimization steps on a given teacher corpus before merging; we run five rounds and evaluate after each merge.

- STD/MTD: We train for 250 steps and save/evaluate checkpoints every 50 steps; the best checkpoint is reported in the main text.

## G.2 STD/MTD AND MoT PER-CHECKPOINT RESULTS

For STD and MTD, we train for 250 steps and save a checkpoint every 50 steps; we evaluate each checkpoint and report the best in the main text.

For MoT, we alternate the base model across the four STD corpora (QWQ, Qwen3-32B, Qwen3-235B, Deepseek-R1), training 50 steps on each corpus and then performing a merge; this constitutes

Table 22: Complete ablations on AIME 2024 (A24) and AIME 2025 (A25). Each entry is a 16-run average. We report per-checkpoint results for STD/MTD (every 50 steps, up to 250), and per-round results for MoT (Rounds 1–5).

| Method | Config | BOBA-200 | | | | | | S1K-200 | | | | | |
| | | Qwen3-8B | | Qwen3-14B | | Qwen3-30B-A3B | | Qwen3-8B | | Qwen3-14B | | Qwen3-30B-A3B | |
| | | A24 | A25 | A24 | A25 | A24 | A25 | A24 | A25 | A24 | A25 | A24 | A25 |
|---|---|---|---|---|---|---|---|---|---|---|---|---|---|
| Base model | (40k) | 75.83 | 67.08 | 79.17 | 70.00 | 81.67 | 72.50 | 75.83 | 67.08 | 79.17 | 70.00 | 81.67 | 72.50 |
| STD (Qwen3-32B) | STEP 50 | 75.42 | 67.71 | 77.71 | 71.25 | 81.04 | 76.04 | 77.50 | 66.67 | 79.79 | 72.50 | 79.58 | 73.13 |
| | STEP 100 | 74.17 | 65.83 | 77.71 | 68.13 | 80.83 | 72.50 | 74.58 | 68.96 | 77.71 | 70.21 | 79.58 | 70.63 |
| | STEP 150 | 75.41 | 63.96 | 78.13 | 66.04 | 81.88 | 72.92 | 73.75 | 67.71 | 79.58 | 72.08 | 80.63 | 70.42 |
| | STEP 200 | 74.58 | 63.75 | 76.67 | 66.88 | 80.63 | 75.63 | 75.21 | 66.67 | 79.79 | 69.58 | 79.58 | 70.83 |
| | STEP 250 | 73.96 | 62.92 | 77.50 | 70.21 | 79.38 | 69.79 | 76.04 | 66.04 | 77.29 | 70.63 | 79.17 | 70.00 |
| STD (Qwen3-235B) | STEP 50 | 74.58 | 67.92 | 78.13 | 74.79 | 80.00 | 78.13 | 74.38 | 68.54 | 77.92 | 72.71 | 77.92 | 75.63 |
| | STEP 100 | 73.13 | 68.33 | 79.17 | 74.79 | 81.88 | 75.42 | 72.50 | 65.83 | 77.08 | 75.41 | 77.08 | 76.88 |
| | STEP 150 | 71.88 | 66.67 | 78.13 | 70.42 | 77.92 | 76.04 | 74.17 | 67.71 | 77.71 | 72.08 | 78.54 | 74.58 |
| | STEP 200 | 71.04 | 65.83 | 77.29 | 74.17 | 79.58 | 75.83 | 71.46 | 67.29 | 78.75 | 73.13 | 78.33 | 74.58 |
| | STEP 250 | 75.00 | 67.29 | 79.38 | 74.17 | 80.42 | 73.54 | 73.96 | 67.08 | 76.67 | 71.46 | 79.17 | 76.04 |
| STD (QWQ) | STEP 50 | 72.50 | 64.38 | 76.46 | 68.54 | 79.58 | 72.50 | 73.53 | 69.17 | 79.17 | 73.54 | 80.83 | 72.08 |
| | STEP 100 | 75.00 | 67.08 | 78.33 | 73.33 | 78.54 | 76.46 | 76.04 | 68.13 | 79.58 | 71.88 | 81.46 | 72.92 |
| | STEP 150 | 75.21 | 67.29 | 79.58 | 73.54 | 79.79 | 75.63 | 75.21 | 65.42 | 79.17 | 73.33 | 80.63 | 68.96 |
| | STEP 200 | 75.83 | 65.83 | 77.29 | 71.46 | 78.54 | 73.96 | 74.58 | 65.63 | 80.21 | 72.92 | 82.08 | 70.63 |
| | STEP 250 | 76.25 | 67.50 | 78.54 | 71.67 | 78.33 | 75.83 | 74.58 | 64.17 | 77.92 | 74.79 | 81.25 | 70.83 |
| STD (Deepseek-R1) | STEP 50 | 67.71 | 60.21 | 74.38 | 67.50 | 78.33 | 68.96 | 70.00 | 61.46 | 73.75 | 62.92 | 78.54 | 70.63 |
| | STEP 100 | 70.21 | 53.33 | 73.75 | 63.33 | 75.00 | 69.79 | 68.54 | 58.33 | 73.33 | 63.13 | 76.46 | 64.58 |
| | STEP 150 | 65.83 | 56.04 | 74.58 | 63.75 | 74.79 | 64.38 | 67.92 | 52.08 | 73.96 | 62.71 | 75.63 | 66.04 |
| | STEP 200 | 65.21 | 53.75 | 74.58 | 64.79 | 74.58 | 67.50 | 66.67 | 55.83 | 71.88 | 61.25 | 74.17 | 65.21 |
| | STEP 250 | 66.67 | 55.42 | 72.50 | 63.54 | 75.42 | 66.88 | 66.88 | 51.67 | 72.71 | 63.96 | 74.17 | 70.00 |
| MTD (ALL TEACHERS) | STEP 50 | 68.54 | 61.04 | 74.79 | 66.88 | 79.17 | 72.92 | 70.83 | 63.54 | 75.83 | 70.83 | 76.46 | 72.08 |
| | STEP 100 | 73.75 | 66.46 | 76.88 | 72.92 | 79.17 | 73.75 | 75.63 | 70.83 | 78.75 | 73.13 | 77.29 | 75.42 |
| | STEP 150 | 71.88 | 68.64 | 76.46 | 75.42 | 77.92 | 72.92 | 73.33 | 66.88 | 79.17 | 73.34 | 78.33 | 74.58 |
| | STEP 200 | 75.00 | 66.04 | 79.58 | 72.50 | 78.75 | 73.75 | 74.17 | 69.38 | 77.08 | 73.33 | 78.33 | 74.58 |
| | STEP 250 | 76.04 | 68.96 | 77.29 | 73.54 | 79.38 | 73.96 | 73.96 | 69.17 | 79.79 | 73.13 | 79.58 | 72.71 |
| MoT (ours) | Round 1 | 72.29 | 66.88 | 78.75 | 73.95 | 80.63 | 73.13 | 74.79 | 69.17 | 78.33 | 69.79 | 80.00 | 75.42 |
| | Round 2 | 75.83 | 69.79 | 79.58 | 73.54 | 79.79 | 76.04 | 77.71 | 70.63 | 80.21 | 74.38 | 82.29 | 74.58 |
| | Round 3 | 76.67 | 70.42 | 80.00 | 74.79 | 80.00 | 77.92 | 76.25 | 70.00 | 80.00 | 74.38 | 79.79 | 74.79 |
| | Round 4 | 78.33 | 70.63 | 79.38 | 76.88 | 81.25 | 75.63 | 77.50 | 71.67 | 79.38 | 75.00 | 80.83 | 77.50 |
| | Round 5 | 76.45 | 66.88 | 78.96 | 73.75 | 82.92 | 78.33 | 76.25 | 68.13 | 81.67 | 75.63 | 80.00 | 77.50 |

one merge round. We run five rounds in total and evaluate after every round. The complete per-round results for all base models and both sources (BOBA-200 and S1K-200) are reported in Table 22.

Key observations from the ablations:

1. MoT consistently yields the strongest distillation gains in almost all settings.

2. For 8B/14B bases, MTD typically surpasses the best single-teacher STD, indicating beneficial complementarity across teachers.

3. For 30B-A3B, MTD brings little to no gain. We hypothesize that QWQ, Qwen3-32B, and Deepseek-R1 are not clearly stronger than the 30B base, so the union is dominated by Qwen3-235B; in contrast, MoT can glean useful signals from the other teachers while mitigating noise, yielding the best results.

## G.3 DETAILED MoT (WITHOUT R1) RESULTS ON BOBA-200

Table 23 reports per-round AIME scores for MoT after ablating the Deepseek-R1 teacher (all other settings identical). AVG is computed as the mean of AIME24 and AIME25.

Table 23: MoT without Deepseek-R1 on BOBA-200: per-round AIME24/AIME25 and AVG. AVG = (AIME24 + AIME25)/2.

| Base model | Round | AIME24 | AIME25 | AVG |
|---|---|---|---|---|
| Qwen3-8B | Round 1 | 75.21 | 69.17 | 72.19 |
| | Round 2 | 75.42 | 72.29 | 73.86 |
| | Round 3 | 76.67 | 70.00 | 73.34 |
| | Round 4 | 78.13 | 69.17 | 73.65 |
| | Round 5 | 76.46 | 69.79 | 73.13 |
| Qwen3-14B | Round 1 | 80.63 | 72.71 | 76.67 |
| | Round 2 | 79.79 | 74.58 | 77.19 |
| | Round 3 | 80.83 | 74.58 | 77.71 |
| | Round 4 | 81.04 | 74.79 | 77.92 |
| | Round 5 | 79.58 | 74.79 | 77.19 |
| Qwen3-30B | Round 1 | 81.88 | 75.00 | 78.44 |
| | Round 2 | 81.88 | 77.08 | 79.48 |
| | Round 3 | 81.25 | 78.75 | 80.00 |
| | Round 4 | 81.88 | 77.71 | 79.80 |
| | Round 5 | 80.42 | 80.00 | 80.21 |

Overall, while MoT without R1 remains competitive, the best AVG per model is consistently below the corresponding full MoT results reported in the main text. This supports the claim that R1 offers complementary supervision that raises the training ceiling and improves late-stage generalization.

## G.4 DETAILED MoT WITH PEER-LEVEL TEACHERS (QWQ + QWEN3-32B) ON BOBA-200

We find that teacher usefulness is not limited to strictly stronger models. Although QWQ, Qwen3-32B, and Qwen3-30B-A3B have comparable parameter scale adn reasoning performance, distilling Qwen3-30B-A3B from peer-level teachers (QWQ or Qwen3-32B) still yields gains. This might imply that what truly benefits the model is not necessarily higher-quality reasoning trajectories, and reasoning trajectories distilled from peer-level teachers can still help. In addition, combining peer-level heterogeneous trajectories with MoT further improves results, and using all teachers performs best. Table 24 reports 16-run AIME averages on BOBA-200 with Qwen3-30B-A3B as the base. Table 25 reports per-round AIME scores for MoT when using only peer-level teachers (QWQ and Qwen3-32B) with Qwen3-30B as the base. AVG is computed as the mean of AIME24 and AIME25.

Overall, these findings support two key conclusions:

(1) Reasoning trajectories distilled from peer-level teachers can still help.

(2) MoT robustly integrates complementary and even distribution-shifted supervision, extracting useful signals while mitigating noise.

Table 24: Peer-level teachers can still help. Results on BOBA-200 with Qwen3-30B-A3B as the base; AIME scores are 16-run averages, AVG is the mean of AIME24 and AIME25.

| Teacher setting | AIME24 | AIME25 | AVG |
|---|---|---|---|
| Base | 80.63 | 70.00 | 75.32 |
| STD: only QWQ | 79.79 | 75.63 | 77.71 |
| STD: only Qwen3-32B | 81.04 | 76.04 | 78.54 |
| MoT: QWQ + Qwen3-32B | 81.04 | 77.29 | 79.17 |
| MoT: ALL TEACHERS | 82.92 | 78.33 | 80.63 |

Table 25: MoT with peer-level teachers (QWQ + Qwen3-32B) on BOBA-200: per-round AIME24/AIME25 and AVG for Qwen3-30B. AVG = (AIME24 + AIME25)/2.

| Round | AIME24 | AIME25 | AVG |
|---|---|---|---|
| Round 1 | 82.70 | 73.95 | 78.33 |
| Round 2 | 80.83 | 74.58 | 77.71 |
| Round 3 | 82.08 | 75.83 | 78.96 |
| Round 4 | 80.83 | 75.00 | 77.92 |
| Round 5 | 81.04 | 77.29 | 79.17 |

# H  BENCHMARK CATEGORIES AND DETAILS

We evaluate nine benchmarks under three categories—**catastrophic-forgetting–sensitive**, **reasoning–knowledge**, and **pure reasoning**—to assess whether CoT-style training with MoT preserves basic capabilities while strengthening reasoning. Here we have provided detailed content and descriptions of these tasks, and given the MoTivations for using them for evaluation and classifying them into the corresponding task categories.

## H.1  CATASTROPHIC-FORGETTING–SENSITIVE TASKS

**CEVAL (CEV).**

*Description:* CEVAL is a Chinese multi-discipline multiple-choice exam suite with approximately 14,000 items spanning 52 subjects at varying difficulty levels.

*Task:* It evaluates factual and domain knowledge recall across humanities, sciences, and professional tracks.

*MoTivation:* It probes retention of broad multilingual knowledge that can degrade after CoT-style training.

**SUPER_GPQA (SG).**

*Description:* SUPER_GPQA is a graduate-level, multi-domain multiple-choice benchmark covering a wide range of academic disciplines.

*Task:* It measures advanced factual knowledge with light multi-step reasoning.

*MoTivation:* It tests whether extensive pretraining knowledge is preserved following CoT fine-tuning.

**IFEVAL (IFE).**

*Description:* IFEVAL is an instruction-following suite with automatically verifiable constraints such as length, formatting, and keyword usage.

*Task:* It evaluates instruction compliance and adherence to explicit constraints.

*MoTivation:* It checks for forgetting of fundamental alignment and compliance behaviors after CoT training.

## H.2  REASONING–KNOWLEDGE TASKS

**SIMPLE_QA (SQ).**

*Description:* SIMPLE_QA is a collection of short, unambiguous fact-seeking questions with a single correct answer. *Task:* It evaluates factual accuracy and calibrated answering by discouraging uninformed guessing.

*MoTivation:* It tests whether CoT improves precision while avoiding hallucinations or overconfident errors.

**MMLU_PRO (MP).**

*Description:* MMLU_PRO is a harder variant of MMLU that increases item difficulty and option counts to emphasize reasoning.

*Task:* It measures multi-step reasoning grounded in broad subject knowledge across many domains.

*MoTivation:* It assesses whether CoT enhances reasoning while maintaining robust domain knowledge.

**MMLU_REDUX (MR).**

*Description:* MMLU_REDUX is a curated and corrected subset of MMLU designed to reduce labeling noise.

*Task:* It evaluates multi-subject knowledge with some analytical reasoning under cleaner annotations.

*MoTivation:* It isolates capability changes from dataset artifacts and checks knowledge retention under CoT.

### H.3 PURE REASONING TASKS

**PhyBench (PB).**

*Description:* PhyBench is a set of physics problems ranging from high-school to Olympiad level that require careful quantitative reasoning.

*Task:* It measures multi-step physics reasoning including derivations and the coordination of multiple principles.

*MoTivation:* It emphasizes chain-of-thought style reasoning rather than rote memorization of facts.

**LiveCodeBench (LCB).**

*Description:* LiveCodeBench is a contamination-controlled suite of recent competitive programming problems drawn from diverse sources.

*Task:* It evaluates algorithmic reasoning, program synthesis, and debugging under executable tests.

*MoTivation:* It probes problem decomposition and step-by-step solution planning independent of encyclopedic knowledge.

**GPQA-Diamond (GPQA-D).**

*Description:* GPQA-Diamond is the hardest expert-vetted subset of GPQA spanning biology, physics, and chemistry.

*Task:* It measures deep scientific reasoning on challenging multiple-choice questions that resist superficial lookup.

*MoTivation:* It stresses genuine multi-step reasoning and scientific insight rather than retrieval of surface facts.

**Figure 7. Tokens marked with confidence drops in MoT under R1-distilled supervision. Marked tokens cluster on teacher-specific stylistic tokens while derivational tokens stay high. This indicates that MoT attenuates inter-teacher inductive biases while preserving consensus reasoning steps. (Note: only tokens after <|im_start|>are included in the loss.)**

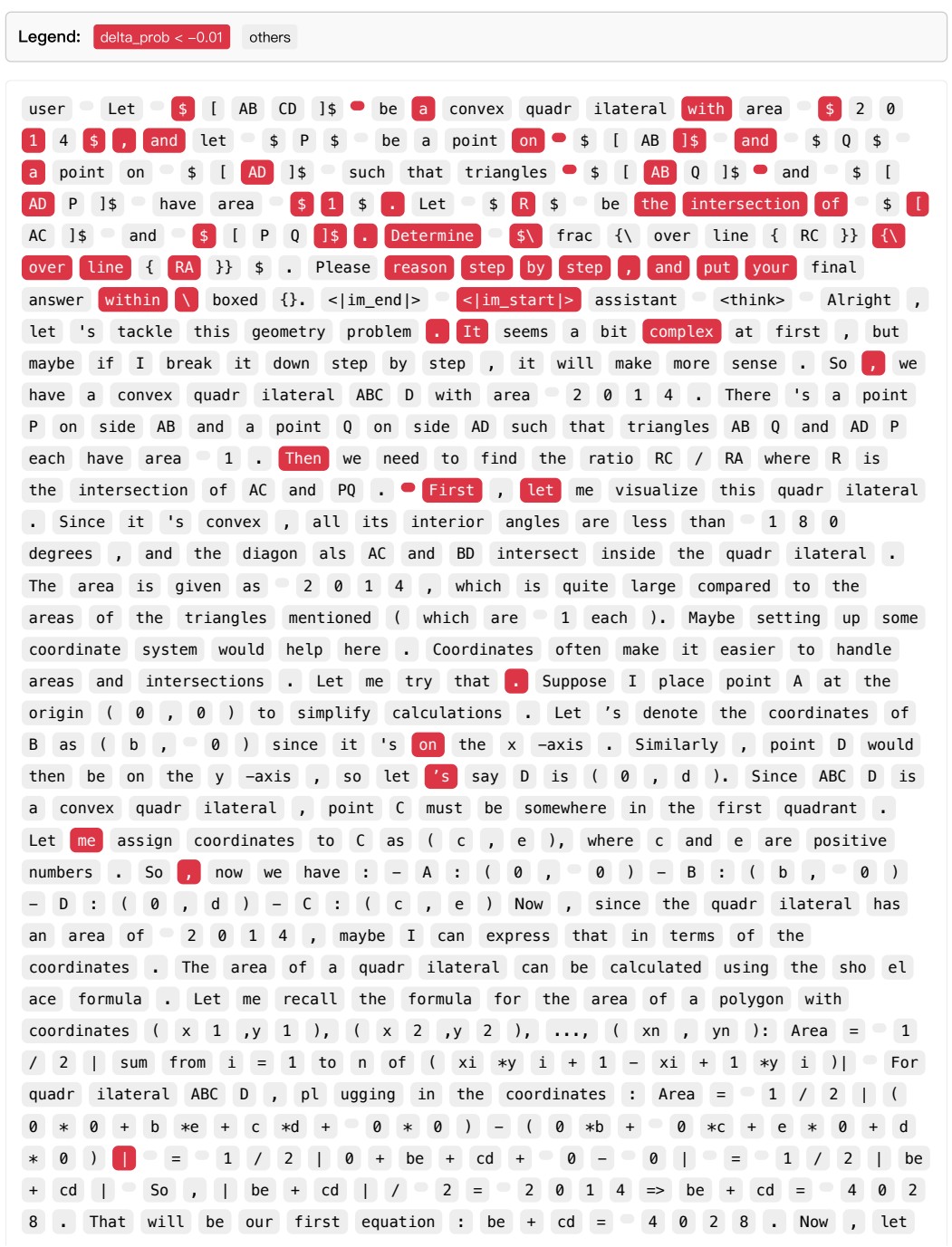

**Figure 7. Tokens marked with confidence drops in MoT under R1-distilled supervision. Marked tokens cluster on teacher-specific stylistic tokens while derivational tokens stay high. This indicates that MoT attenuates inter-teacher inductive biases while preserving consensus reasoning steps. (Note: only tokens after $<|$im_start$|>$ are included in the loss.)**

's consider points P and Q . Point P is on AB , and point Q is on AD . So , param etrize their coordinates . Since AB is from ( 0 , 0 ) to ( b , 0 ), a point P on AB can be **represented** as ( p , 0 ) where p is between 0 and b . Similarly , AD goes from ( 0 , 0 ) to ( 0 ,d ), so Q can be represented as ( 0 , q ) where q is between 0 and d . Now , the problem states that triangles AB Q and AD P have area 1 each . First , triangle AB Q : points A , B , Q . The coordinates are A ( 0 , 0 ), B (b , 0 ), Q ( 0 ,q ). The area can be calculated as 1 / 2 * base * height . The base AB has length b , and the height is the distance from Q to AB , which is q ( since AB is along the x −axis and Q is on the y −axis ). So area is ( 1 / 2 )* b *q = 1 . Hence , ( 1 / 2 )* b *q = 1 => **b** *q = 2 . Equation 2 : bq = 2 . Similarly , triangle AD P : points A , D , P . Coordinates A ( 0 , 0 ), D ( 0 ,d ), P (p , 0 ). The area can again be calculated as 1 / 2 * base * height . **The** base AD is length d , and the height is the **distance** from P to AD , which is p ( since AD is along the y −axis and P is on the x −axis ). So area is ( 1 / 2 )* d *p = 1 => d *p = 2 . Equation 3 : dp = 2 . So **,** from equations 2 and 3 , we have bq = 2 and dp = 2 . Therefore , we can solve for q and p : q = 2 /b and p = 2 /d . **So** , Q is at ( 0 , 2 /b ) and P is at ( 2 /d , 0 ). **Now** , we need to find the intersection point R of AC and PQ . First , let 's write equations for lines AC and PQ . Line AC connects points A ( 0 , 0 ) and C (c ,e ). **So** , the param etric equations for AC can be written as x = c *t , y = e *t , where t is in [ 0 , 1 **].** Line PQ connects points P ( 2 /d , 0 ) and Q ( 0 , 2 /b ). To find the equation of PQ , let 's compute the slope first . Slo pe = ( 2 /b − 0 )/( 0 − 2 /d ) = ( 2 /b )/ (− 2 /d ) = − d /b . So , the slope of PQ is − d /b . Therefore , the equation of PQ can be written in point −s lope form . Using point P ( 2 /d , 0 ): y − 0 = (− d /b )( x − 2 /d ) => y = (− d /b )x + ( d /b )( 2 /d ) => y = (− d /b )x + 2 /b . Alternatively , using Q : y − 2 /b = (− d /b )( x − 0 ) => y = (− d /b )x + 2 /b . Same equation . So , equation of PQ : y = (− d /b )x + 2 /b . Now , we need to find the intersection point R of AC and PQ . Param etr izing AC as ( ct , et ). Substitute into PQ 's equation : et = (− d /b )( ct ) + 2 /b . So , solve for t : et = (− cd /b )t + 2 /b et + ( cd /b )t = 2 /b t (e + cd /b ) = 2 /b t = ( 2 /b ) / ( e + cd /b ) = ( 2 /b ) / (( be + cd )/ b ) **=** ( 2 /b ) * ( b / ( be + cd )) = 2 / ( be + cd ). But earlier , we found that be + cd = 4 0 2 8 . Therefore **,** t = 2 / 4 0 2 8 =

**Figure 7. Tokens marked with confidence drops in MoT under R1-distilled supervision. Marked tokens cluster on teacher-specific stylistic tokens while derivational tokens stay high. This indicates that MoT attenuates inter-teacher inductive biases while preserving consensus reasoning steps. (Note: only tokens after $<|im\_start|>$ are included in the loss.)**

1 / 2 0 1 4 . Therefore , the coordinates of R are ( c * t , e * t ) = ( c / 2 0 1 4 , e / 2 0 1 4 ). So , R divides AC such that RA is the distance from A to R , and RC is from R to C . Since AC is parameter ized by t from 0 to 1 , with t = 0 at A and t = 1 at C , the ratio RC / RA is ( 1 – t )/ t . But in this case , t is 1 / 2 0 1 4 . Therefore , RC / RA = ( 1 – 1 / 2 0 1 4 ) / ( 1 / 2 0 1 4 ) ) = ( 2 0 1 3 / 2 0 1 4 ) / ( 1 / 2 0 1 4 ) ) = 2 0 1 3 . H ence , the ratio is 2 0 1 3 . Wait , so the answer is 2 0 1 3 ? That seems plausible . Let me check my steps again to be sure . 1 . Set coordinates for the quadr ilateral with A at ( 0 , 0 ), B ( b , 0 ), D ( 0 ,d ), C ( c ,e ). Correct . 2 . Calcul ated area using sho el ace formula : got be + cd = 4 0 2 8 . Seems right . 3 . Found points P ( 2 /d , 0 ) and Q ( 0 , 2 /b ). Yes , based on the area conditions . 4 . Equation for PQ : derived slope – d /b and equation y = (– d /b )x + 2 /b . Check . 5 . Param etr izing AC as ( ct , et ) and solving for intersection . Found t = 1 / 2 0 1 4 . Then , RA is the distance from A to R : since R is at t = 1 / 2 0 1 4 , RA corresponds to t = 1 / 2 0 1 4 , and RC is the remaining distance from R to C : 1 – t . Therefore , ratio RC / RA = ( 1 – t )/ t = ( 1 – 1 / 2 0 1 4 )/( 1 / 2 0 1 4 ) = ( 2 0 1 3 / 2 0 1 4 )/( 1 / 2 0 1 4 ) = 2 0 1 3 . So yes , seems correct . Alternatively , since coordinates are linear , t is the parameter along AC , so the ratios are related directly to the parameter t . So another way : RA : RC = t : ( 1 – t ), so RC / RA = ( 1 – t )/ t = ( 1 /( t ) ) – 1 . Since t = 1 / 2 0 1 4 , RC / RA = 2 0 1 4 – 1 = 2 0 1 3 . Perfect . Therefore , the final answer is 2 0 1 3 . Pretty elegant ! Using coordinates made it straightforward , even if setting up the coordinates took some time . Let me just think if there 's any other way without coordinates , but perhaps this is the most direct method here . ** Final Answer ** \ boxed { 2 0 1 3 } </think> Given a convex quadr ilateral \( AB CD \) with area 2 0 1 4 , points \( P \) on \( AB \) and \( Q \) on \( AD \) are such that triangles \( AB Q \) and \( AD P \) each have area 1 . We need to determine the ratio \ (\ frac {\ over line { RC }} {\ over line { RA }} \) where \( R \) is the intersection of \( AC \) and \( P Q \ ). 1 . ** Coordinate Setup ** : – Place \( A \) at \ (( 0 , 0 )\ ), \( B \) at \ (( b , 0 )\ ), \( D \) at \ (( 0 , d )\ ), and \( C \) at \ (( c , e )\ ). – The area of \( AB CD \) is given by the sho el ace formula , leading to the equation \( be + cd = 4 0 2 8 \ ). 2 . ** Points \( P \) and \( Q \ )** : – \( P \) on

**Figure 7. Tokens marked with confidence drops in MoT under R1-distilled supervision. Marked tokens cluster on teacher-specific stylistic tokens while derivational tokens stay high. This indicates that MoT attenuates inter-teacher inductive biases while preserving consensus reasoning steps. (Note: only tokens after <|im_start|> are included in the loss.)**

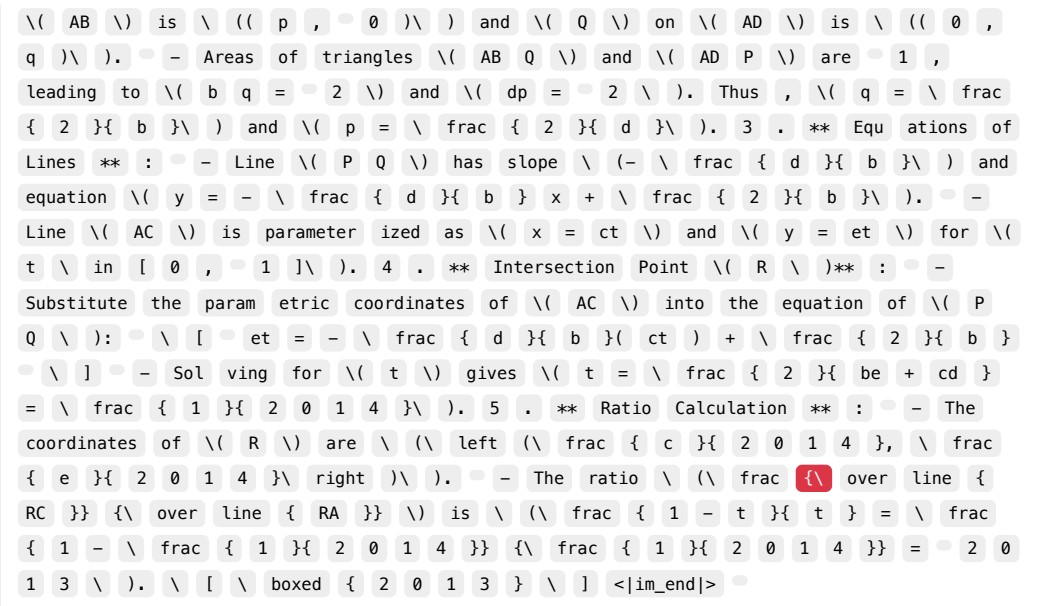

**Figure 8. Tokens marked with confidence drops in STD(R1) under R1-distilled supervision. Marked tokens are rare to nearly absent, indicating that direct distillation fully adopts the teacher's trajectory, including stylistic tokens and latent inductive biases, rather than filtering them. (Note: only tokens after <|im_start|>are included in the loss.)**

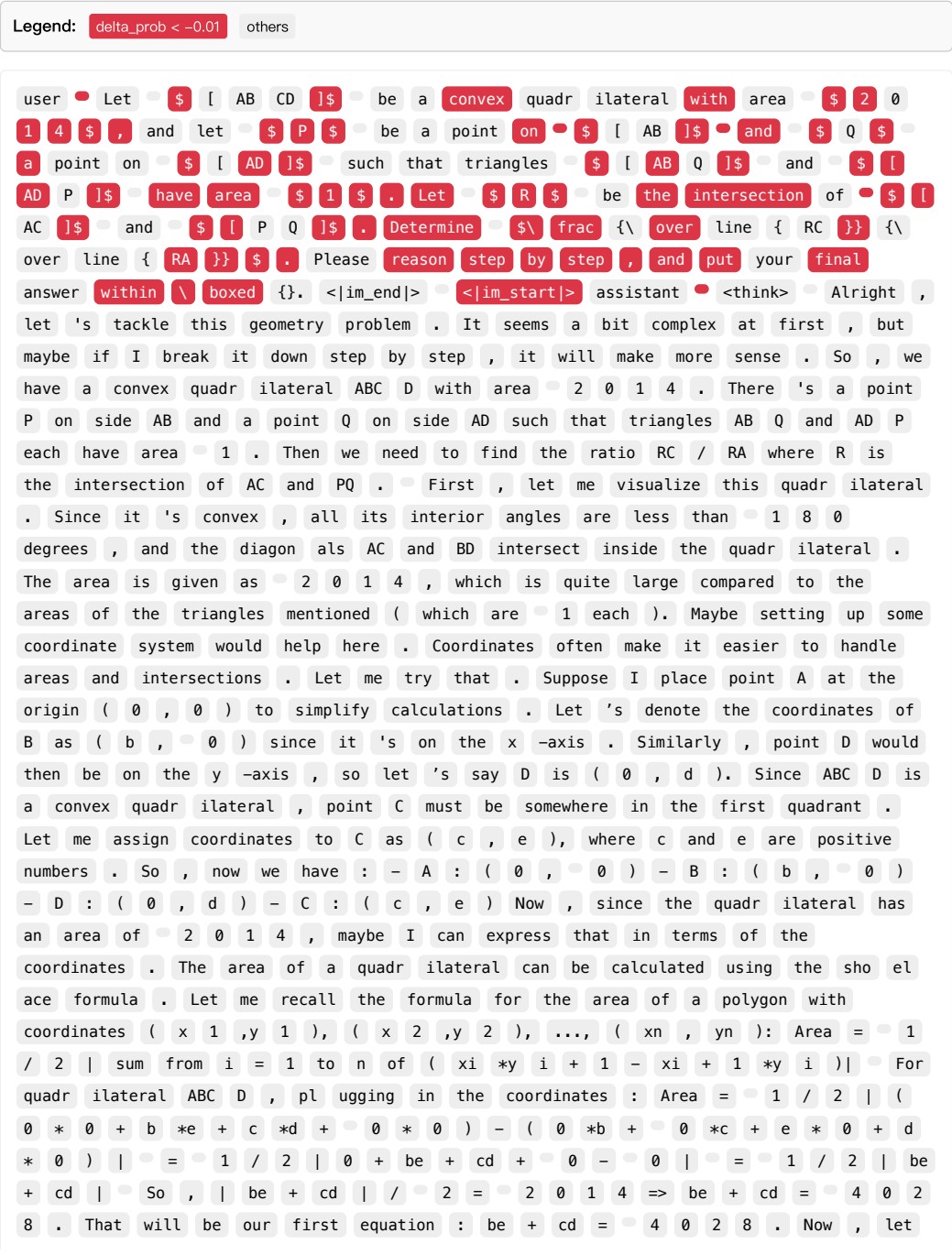

Legend: delta_prob < −0.01    others

user Let $ [ AB CD ]$ be a convex quadr ilateral with area $ 2 0 1 4 $ , and let $ P $ be a point on $ [ AB ]$ and $ Q $ a point on $ [ AD ]$ such that triangles $ [ AB Q ]$ and $ [ AD P ]$ have area $ 1 $ . Let $ R $ be the intersection of $ [ AC ]$ and $ [ P Q ]$ . Determine $\ frac {\ over line { RC }} {\ over line { RA }} $ . Please reason step by step , and put your final answer within \ boxed {}. <|im_end|> <|im_start|> assistant <think> Alright , let 's tackle this geometry problem . It seems a bit complex at first , but maybe if I break it down step by step , it will make more sense . So , we have a convex quadr ilateral ABC D with area 2 0 1 4 . There 's a point P on side AB and a point Q on side AD such that triangles AB Q and AD P each have area 1 . Then we need to find the ratio RC / RA where R is the intersection of AC and PQ . First , let me visualize this quadr ilateral . Since it 's convex , all its interior angles are less than 1 8 0 degrees , and the diagon als AC and BD intersect inside the quadr ilateral . The area is given as 2 0 1 4 , which is quite large compared to the areas of the triangles mentioned ( which are 1 each ). Maybe setting up some coordinate system would help here . Coordinates often make it easier to handle areas and intersections . Let me try that . Suppose I place point A at the origin ( 0 , 0 ) to simplify calculations . Let 's denote the coordinates of B as ( b , 0 ) since it 's on the x −axis . Similarly , point D would then be on the y −axis , so let 's say D is ( 0 , d ). Since ABC D is a convex quadr ilateral , point C must be somewhere in the first quadrant . Let me assign coordinates to C as ( c , e ), where c and e are positive numbers . So , now we have : − A : ( 0 , 0 ) − B : ( b , 0 ) − D : ( 0 , d ) − C : ( c , e ) Now , since the quadr ilateral has an area of 2 0 1 4 , maybe I can express that in terms of the coordinates . The area of a quadr ilateral can be calculated using the sho el ace formula . Let me recall the formula for the area of a polygon with coordinates ( x 1 ,y 1 ), ( x 2 ,y 2 ), ..., ( xn , yn ): Area = 1 / 2 | sum from i = 1 to n of ( xi *y i + 1 − xi + 1 *y i )| For quadr ilateral ABC D , pl ugging in the coordinates : Area = 1 / 2 | ( 0 * 0 + b *e + c *d + 0 * 0 ) − ( 0 *b + 0 *c + e * 0 + d * 0 ) | = 1 / 2 | 0 + be + cd + 0 − 0 | = 1 / 2 | be + cd | So , | be + cd | / 2 = 2 0 1 4 => be + cd = 4 0 2 8 . That will be our first equation : be + cd = 4 0 2 8 . Now , let

**Figure 8. Tokens marked with confidence drops in STD(R1) under R1-distilled supervision. Marked tokens are rare to nearly absent, indicating that direct distillation fully adopts the teacher's trajectory, including stylistic tokens and latent inductive biases, rather than filtering them. (Note: only tokens after <|im_start|>are included in the loss.)**

```
's consider points P and Q . Point P is on AB , and point Q is on AD .
So , param etrize their coordinates . Since AB is from ( 0 , 0 ) to ( b
, 0 ), a point P on AB can be represented as ( p , 0 ) where p is
between 0 and b . Similarly , AD goes from ( 0 , 0 ) to ( 0 ,d ), so
Q can be represented as ( 0 , q ) where q is between 0 and d . Now ,
the problem states that triangles AB Q and AD P have area 1 each . First
, triangle AB Q : points A , B , Q . The coordinates are A ( 0 , 0 ),
B (b , 0 ), Q ( 0 ,q ). The area can be calculated as 1 / 2 * base
* height . The base AB has length b , and the height is the distance from
Q to AB , which is q ( since AB is along the x −axis and Q is on the y
−axis ). So area is ( 1 / 2 )* b *q = 1 . Hence , ( 1 / 2 )* b
*q = 1 => b *q = 2 . Equation 2 : bq = 2 . Similarly , triangle
AD P : points A , D , P . Coordinates A ( 0 , 0 ), D ( 0 ,d ), P (p
, 0 ). The area can again be calculated as 1 / 2 * base * height . The
base AD is length d , and the height is the distance from P to AD , which
is p ( since AD is along the y −axis and P is on the x −axis ). So area
is ( 1 / 2 )* d *p = 1 => d *p = 2 . Equation 3 : dp = 2 .
So , from equations 2 and 3 , we have bq = 2 and dp = 2 .
Therefore , we can solve for q and p : q = 2 /b and p = 2 /d . So
, Q is at ( 0 , 2 /b ) and P is at ( 2 /d , 0 ). Now , we need
to find the intersection point R of AC and PQ . First , let 's write
equations for lines AC and PQ . Line AC connects points A ( 0 , 0 ) and C
(c ,e ). So , the param etric equations for AC can be written as x = c *t
, y = e *t , where t is in [ 0 , 1 ]. Line PQ connects points P ( 2
/d , 0 ) and Q ( 0 , 2 /b ). To find the equation of PQ , let 's
compute the slope first . Slo pe = ( 2 /b − 0 )/( 0 − 2 /d ) = (
2 /b )/ (− 2 /d ) = − d /b . So , the slope of PQ is − d /b .
Therefore , the equation of PQ can be written in point −s lope form . Using
point P ( 2 /d , 0 ): y − 0 = (− d /b )( x − 2 /d ) => y =
(− d /b )x + ( d /b )( 2 /d ) => y = (− d /b )x + 2 /b .
Alternatively , using Q : y − 2 /b = (− d /b )( x − 0 ) => y = (−
d /b )x + 2 /b . Same equation . So , equation of PQ : y = (− d /b
)x + 2 /b . Now , we need to find the intersection point R of AC and PQ
. Param etr izing AC as ( ct , et ). Substitute into PQ 's equation : et
= (− d /b )( ct ) + 2 /b . So , solve for t : et = (− cd /b )t +
2 /b et + ( cd /b )t = 2 /b t (e + cd /b ) = 2 /b t =
( 2 /b ) / ( e + cd /b ) = ( 2 /b ) / (( be + cd )/ b ) = ( 2
/b ) * ( b / ( be + cd )) = 2 / ( be + cd ). But earlier , we
found that be + cd = 4 0 2 8 . Therefore , t = 2 / 4 0 2 8 =
```

**Figure 8. Tokens marked with confidence drops in STD(R1) under R1-distilled supervision. Marked tokens are rare to nearly absent, indicating that direct distillation fully adopts the teacher's trajectory, including stylistic tokens and latent inductive biases, rather than filtering them. (Note: only tokens after $<$|im_start|$>$ are included in the loss.)**

1 / 2 0 1 4 . Therefore , the coordinates of R are ( c * t , e * t ) = ( c / 2 0 1 4 , e / 2 0 1 4 ). So , R divides AC such that RA is the distance from A to R , and RC is from R to C . Since AC is parameter ized by t from 0 to 1 , with t = 0 at A and t = 1 at C , the ratio RC / RA is ( 1 – t )/ t . But in this case , t is 1 / 2 0 1 4 . Therefore , RC / RA = ( 1 – 1 / 2 0 1 4 ) / ( 1 / 2 0 1 4 ) ) = ( 2 0 1 3 / 2 0 1 4 ) / ( 1 / 2 0 1 4 ) ) = 2 0 1 3 . H ence , the ratio is 2 0 1 3 . Wait , so the answer is 2 0 1 3 ? That seems plausible . Let me check my steps again to be sure . 1 . Set coordinates for the quadr ilateral with A at ( 0 , 0 ), B (b , 0 ), D ( 0 ,d ), C (c ,e ). Correct . 2 . Calcul ated area using sho el ace formula : got be + cd = 4 0 2 8 . Seems right . 3 . Found points P ( 2 /d , 0 ) and Q ( 0 , 2 /b ). Yes , based on the area conditions . 4 . Equation for PQ : derived slope – d /b and equation y = (– d /b )x + 2 /b . Check . 5 . Param etr izing AC as ( ct , et ) and solving for intersection . Found t = 1 / 2 0 1 4 . Then , RA is the distance from A to R : since R is at t = 1 / 2 0 1 4 , RA corresponds to t = 1 / 2 0 1 4 , and RC is the remaining distance from R to C : 1 – t . Therefore , ratio RC / RA = ( 1 – t )/ t = ( 1 – 1 / 2 0 1 4 )/( 1 / 2 0 1 4 ) = ( 2 0 1 3 / 2 0 1 4 )/( 1 / 2 0 1 4 ) = 2 0 1 3 . So yes , seems correct . Alternatively , since coordinates are linear , t is the parameter along AC , so the ratios are related directly to the parameter t . So another way : RA : RC = t : ( 1 – t ), so RC / RA = ( 1 – t )/ t = ( 1 /( t ) ) – 1 . Since t = 1 / 2 0 1 4 , RC / RA = 2 0 1 4 – 1 = 2 0 1 3 . Perfect . Therefore , the final answer is 2 0 1 3 . Pretty elegant ! Using coordinates made it straightforward , even if setting up the coordinates took some time . Let me just think if there 's any other way without coordinates , but perhaps this is the most direct method here . ** Final Answer ** \ boxed { 2 0 1 3 } </think> Given a convex quadr ilateral \( AB CD \) with area 2 0 1 4 , points \( P \) on \( AB \) and \( Q \) on \( AD \) are such that triangles \( AB Q \) and \( AD P \) each have area 1 . We need to determine the ratio \ (\ frac {\ over line { RC }} {\ over line { RA }} \) where \( R \) is the intersection of \( AC \) and \( P Q \ ). 1 . ** Coordinate Setup ** : – Place \( A \) at \ (( 0 , 0 )\ ), \( B \) at \ (( b , 0 )\ ), \( D \) at \ (( 0 , d )\ ), and \( C \) at \ (( c , e )\ ). – The area of \( AB CD \) is given by the sho el ace formula , leading to the equation \( be + cd = 4 0 2 8 \ ). 2 . ** Points \( P \) and \( Q \ )** : – \( P \) on

**Figure 8. Tokens marked with confidence drops in STD(R1) under R1-distilled supervision. Marked tokens are rare to nearly absent, indicating that direct distillation fully adopts the teacher's trajectory, including stylistic tokens and latent inductive biases, rather than filtering them. (Note: only tokens after $<|\text{im\_start}|>$ are included in the loss.)**

\( AB \) is \ (( p , 0 )\ ) and \( Q \) on \( AD \) is \ (( 0 , q )\ ). − Areas of triangles \( AB Q \) and \( AD P \) are 1 , leading to \( b q = 2 \) and \( dp = 2 \ ). Thus , \( q = \ frac { 2 }{ b }\ ) and \( p = \ frac { 2 }{ d }\ ). 3 . ** Equ ations of Lines ** : − Line \( P Q \) has slope \ (− \ frac { d }{ b }\ ) and equation \( y = − \ frac { d }{ b } x + \ frac { 2 }{ b }\ ). − Line \( AC \) is parameter ized as \( x = ct \) and \( y = et \) for \( t \ in [ 0 , 1 ]\ ). 4 . ** Intersection Point \( R \ )** : − Substitute the param etric coordinates of \( AC \) into the equation of \( P Q \ ): \ [ et = − \ frac { d }{ b }( ct ) + \ frac { 2 }{ b } \ ] − Sol ving for \( t \) gives \( t = \ frac { 2 }{ be + cd } = \ frac { 1 }{ 2 0 1 4 }\ ). 5 . ** Ratio Calculation ** : − The coordinates of \( R \) are \ (\ left (\ frac { c }{ 2 0 1 4 }, \ frac { e }{ 2 0 1 4 }\ right )\ ). − The ratio \ (\ frac {\ over line { RC }} {\ over line { RA }} \) is \ (\ frac { 1 − t }{ t } = \ frac { 1 − \ frac { 1 }{ 2 0 1 4 }} {\ frac { 1 }{ 2 0 1 4 }} = 2 0 1 3 \ ). \ [ \ boxed { 2 0 1 3 } \ ] $<|\text{im\_end}|>$

## THE USE OF LARGE LANGUAGE MODELS (LLMS)

We used large language models (LLMs) as general-purpose assist tools for editing (English proof-reading, minor wording/LaTeX refactoring) and for generating figure/table captions drafts that were subsequently verified and rewritten by the authors. LLMs did not design experiments, select results, write the core method, or generate evaluation numbers. All experimental outputs, metrics, and plots derive from our released code and logs. Separately, the research subject of this paper employs teacher LLMs to produce chains-of-thought for distillation; this is part of the method under study, not assistance in authorship. The authors take full responsibility for the content and have verified factual claims and citations. No text was copied from third-party sources without attribution.

