# OpenReview forum: "Merge-of-Thought Distillation"
_ICLR.cc/2026/Conference — Submitted to ICLR 2026_

### Official Review · Reviewer_HWmF · 2025-10-31

**Soundness:** 3
**Presentation:** 3
**Contribution:** 1
**Rating:** 4
**Confidence:** 4

**Summary:**

This paper proposes "Merge-of-Thought Distillation (MoT)", a framework that alternates between teacher-specific branch fine-tuning and weight-space merging to consolidate the chain-of-thought reasoning capabilities of multiple large language models into a single student model. The authors claim that with only about 200 samples, their method can surpass some larger models on mathematical reasoning benchmarks and mitigate catastrophic forgetting.

**Strengths:**

1) It provides a systematic study of multi-teacher long chain-of-thought distillation, addressing a gap in the existing literature which often focuses on single-teacher settings.
2) The experimental design is comprehensive, covering various model scales and datasets, and is supported by thorough ablation studies and theoretical analysis.

**Weaknesses:**

1) The core method of "branch training and weight averaging" is essentially a straightforward application of existing model merging techniques, lacking substantial algorithmic or structural innovation. The overarching idea of multi-teacher fusion is not a breakthrough in itself.
2) The paper identifies the critical problem of "teacher selection mattering" but does not address it; instead, the method circumvents it by merging all teachers. It lacks dynamic assessment of teacher quality or a weighted merging mechanism.
3) The core methodology is overly simplistic and does not incorporate more advanced fusion strategies. The experiments fail to adequately demonstrate its generalization capability to non-mathematical tasks, and the trade-off between computational cost and performance gain is not sufficiently discussed.
4) Some baseline results are taken from original papers without reproduction under a unified setup, potentially compromising fairness. The absence of comparisons with more state-of-the-art multi-teacher distillation or fusion methods weakens the persuasiveness of the claims.

**Questions:**

1) The paper compares MoT against naive multi-teacher data mixing (MTD) and one-shot weight merging. However, a more fair baseline would be an ensemble of the multiple independent teacher models, i.e., averaging their output probabilities at inference time. What are the significant advantages of MoT over such an output-level ensemble in terms of performance, efficiency, or stability?
2) Weight averaging is a very basic and sensitive merging technique. Did the authors experiment with more advanced merging algorithms, such as TIES-Merging or Task Arithmetic? If so, what were the results? If not, why was such a simple method chosen as the core innovative component?
3) The paper lacks comparisons with state-of-the-art methods specifically designed for multi-teacher or data fusion. For example, what are the distinct advantages and disadvantages of MoT compared to Gradient Blending (weighting teachers at the gradient level) or DARE (a pruning method before weight merging)?

---

> ### Author Response · Authors · 2025-11-20
> **Author Response to Reviewer HWmF (Part I)**
>
> Thank you for your prompt response. We sincerely appreciate your insightful feedback, which will help us improve our work! We are glad to have the opportunity to discuss and address your concerns to gain your approval of our MoT works.
>
>
> > **Response to Weakness 1 (Novelty of “branch training + averaging” in MoT)**
>
> Thank you for this comment. We fully agree that **“train several branches and average their weights” is not a totally new primitive**. Our contribution lies in **what problem we apply it to, how we motivate it, and what we show about its behavior** in that setting.
>
> 1. **Different problem: unifying conflicting CoTs on the *same* questions.**
>    Existing model-merging work typically studies models trained on **different tasks or domains**, where merging is meant to combine capabilities across data distributions. In our case, all branches are trained on the **same set of questions**, but with different teacher CoTs, which can conflict token-wise even when the final answers agree. The central challenge is therefore not “expanding coverage across tasks,” but reconciling heterogeneous reasoning trajectories and teacher-specific inductive biases on shared problems. Mechanically the merge operation is simple, but conceptually MoT is designed as a procedure for **building consensus reasoning across multiple teachers on the same seed problems**, which, to our knowledge, has not been systematically studied in the long-CoT regime.
>
> 2. **Iterative merging is motivated and validated specifically for multi-teacher long-CoT distillation.**
>    In this setting, we first revisited natural baselines: single-teacher distillation, naive multi-teacher union, and one-shot post-hoc merging. Empirically, these do not reliably outperform the best single teacher and often fail to resolve cross-teacher conflicts. This negative evidence is precisely what motivated our iterative “branch SFT + merge” design: instead of a single merge, we repeatedly (i) let each branch absorb one teacher’s reasoning, then (ii) merge them back to progressively reinforce features that are stable across teachers. Our experiments show that in this multi-teacher long-CoT setting, this very simple procedure is consistently and substantially effectiv.
>
> In this sense, we see the main contribution of MoT not as proposing a new merging primitive, but as **identifying, carefully motivating, and extensively validating a simple merging-based training recipe that is strongly motivated by our prior analysis and proves highly effective for a new and practically important setting: multi-teacher long-CoT distillation**.
>
>
>
>
> > **Response to Question 1 (comparison to output-level teacher ensembles)**
>
> Thank you for pointing out the ensemble baseline. We agree that, conceptually, averaging teacher output probabilities at inference time is a good way to combine multiple models, but in our setting it is very hard to implement:
> 1. **Output-level ensembles are ill-defined for long CoT rationales.**
>    In our regime, different teachers produce different free-form CoTs, and follow **totally different solution paths** even when they end with the same boxed answer. This makes standard probability ensembling (e.g., averaging token-level logits) essentially **undefined in practice**: there is no canonical alignment between the teacher sequences on which to average probabilities. Our MoT procedure is exactly designed to handle this “different CoTs for the same question” setting in **parameter space**, where we can reconcile heterogeneous rationales via branch training + merging, rather than trying to ensemble incompatible token streams.
>
> 2. **Relation to other multi-teacher fusion methods.**
>    In addition to naive data mixing, we have also compared against **strong parameter-space fusion baselines** (TIES, DARE) you mentioned, which can be viewed as more advanced “multi-model fusion” methods. We discussed the detailed results in our responses to other questions below.

---

> ### Author Response · Authors · 2025-11-20
> **Author Response to Reviewer HWmF (Part II)**
>
> > **Response to Weakness 2 Part I (no dynamic teacher-quality assessment )**
>
> Thank you for raising this point. Our goal in MoT is deliberately **selection-free**: instead of explicitly ranking teachers or dynamically reweighting them, we ask whether a very simple, uniform multi-teacher procedure can already avoid brittle teacher selection and still exploit complementary teachers effectively.
>
> However, we agree that explicit, dynamic teacher assessment is an interesting next step, especially for much larger or more heterogeneous teacher pools. Empirically, in our current regime this works surprisingly well. On BOBA-200 with Qwen3-8B, we start from the best single teacher (by STD), then **progressively add** the second-best, third-best, and finally the noisy, distribution-shifted teacher R1 into the MoT pool, keeping everything else fixed:
>
> | # Teachers in MoT | AIME24 | AIME25 | AVG   |
> | ----------------- | ------ | ------ | ----- |
> | 1 (best teacher)  | 76.25  | 67.50  | 71.88 |
> | 2                 | 77.50  | 68.33  | 72.92 |
> | 3                 | 78.13  | 69.17  | 73.65 |
> | 4 (with  R1)      | 78.33  | 70.63  | 74.48 |
>
> We see a monotonic improvement as we add more teachers, and performance continues to increase even after including the strongly shifted / weaker teacher R1. Together with the R1 ablation in Section 7 (where removing R1 hurts MoT), this suggests that **even uniform-weight MoT already behaves as an implicit quality filter**, reinforcing directions that are consistently supported across teachers and rounds, and down-weighting idiosyncratic signals.
>
> We appreciate the reviewer’s insightful comment and have revised the paper accordingly at lines 1024–1055.

---

> ### Author Response · Authors · 2025-11-20
> **Author Response to Reviewer HWmF (Part III)**
>
> > **Response to Weakness 2 Part II (weighted merging mechanism) / Weakness 4 Part II (more state-of-the-art fusion methods) / Question 2 / Question 3**
>
> Thank you for raising the questions about more advanced merging / data-fusion methods and whether our gains come only from using a very simple averaging rule. During the rebuttal period, we explicitly evaluated more advanced merging / fusion algorithms you mentioned in our setting, both as **data-fusion baselines** and as **drop-in replacements for the MoT merge step**.
>
> On **Qwen3-8B + BOBA-200**, we tested:
>
> * **Data-fusion baselines**
>     * **One-shot TIES** merge
>     * **One-shot DARE** merge
>
> * **Replacements for the MoT merge step**
>     * **MoT(TIES)**: MoT where each merge uses TIES instead of averaging
>     * **MoT(DARE)**: MoT where each merge uses DARE instead of averaging
>
> All other settings (teachers, data, schedule, evaluation) are identical. Final AIME AVG:
>
> | Method                       | AVG       |
> | ---------------------------- | --------- |
> | Base                         | 71.46     |
> | One-shot TIES                | 71.67     |
> | One-shot DARE                | 60.42     |
> | **MoT (TIES)**               | 73.34     |
> | **MoT (DARE)**               | 74.17     |
> | **MoT (simple merge)** | **74.48** |
>
> Per-round behavior:
>
> **MoT(TIES)**
>
> | Round | AIME24 | AIME25 | AVG   |
> | ----- | ------ | ------ | ----- |
> | 1     | 73.33  | 66.67  | 70.00 |
> | 2     | 77.50  | 67.50  | 72.50 |
> | 3     | 77.50  | 69.17  | 73.34 |
> | 4     | 74.17  | 65.00  | 69.59 |
> | 5     | 75.00  | 65.83  | 70.42 |
>
> **MoT(DARE)**
>
> | Round | AIME24 | AIME25 | AVG   |
> | ----- | ------ | ------ | ----- |
> | 1     | 75.00  | 62.50  | 68.75 |
> | 2     | 76.67  | 65.00  | 70.84 |
> | 3     | 78.33  | 70.00  | 74.17 |
> | 4     | 79.17  | 69.17  | 74.17 |
> | 5     | 73.33  | 65.83  | 69.58 |
>
> Here, **TIES and DARE serve simultaneously as modern model-merging algorithms and as one-shot multi-teacher / data-fusion baselines**. Three conclusions are clear:
>
> 1. **Advanced one-shot merges alone are not sufficient in our setting.**
>     As **one-shot data-fusion methods**, they are **far weaker than MoT** in our multi-teacher long-CoT setting: one-shot TIES hardly improves over Base, and one-shot DARE actually hurts performance severely. This indicates that more sophisticated methods originally designed for merging models trained on different domains/tasks may be much less directly suitable for **unifying different reasoning paths for the same questions**.
>
> 2. **The iterative MoT process is compatible with and consistently boosts advanced merges.**
>    When we place these same operators(TIES/DARE) **inside the MoT loop**, performance improves substantially, and the DARE's gain of almost **+14 points** is very superising. Algorithmically, this is plausible because one-shot DARE acts on highly conflicting teacher-specific updates all at once, so its aggressive sparsification tends to over-prune useful but partially misaligned reasoning directions and hurts performance. In contrast, **DARE inside MoT sees smaller, progressively more aligned deltas across rounds and branches**, so it behaves like a gradual consensus regularizer that keeps directions repeatedly reinforced by multiple teachers, which explains the large gain when used iteratively. All of this indicates that the key gains indeed come from the iterative merge-and-train procedure.
>
>
> 3. **Simple averaging still achieves the highest and most robust ceiling.**
>    We found that, unlike MoT based on SIMPLE MERGE, TIES and DARE showed faster convergence in the evaluation of across merge round, but seemed to have a slightly lower upper ceiling and much poorer anti-overfitting ability. A plausible explanation, consistent with our Section 7 discussion, is that simple averaging **does not impose any parameter filtering**, allowing MoT to naturally absorb useful signals even from suboptimal/noisy teachers and thereby achieve a higher reasoning ceiling.
>
>
> We have also modified the manuscript to incorporate this thoughtful suggestion, as shown at lines 1000–1024.

---

> ### Author Response · Authors · 2025-11-20
> **Author Response to Reviewer HWmF (Part IV)**
>
> > **Response to Weakness 3 Part I (Generalization beyond math)**
>
> The reviewer’s question about extending MoT beyond mathematical reasoning is well-taken. To directly examine this, we ran MoT on a **code reasoning** domain using **178 code-domain examples** from [1], applying **exactly the same distillation, training, and evaluation pipeline** as in the main paper (same teacher pool, 1Q–multiA CoTs, same MoT procedure). We evaluated on LiveCodeBench and obtained:
>
> | Method         | LiveCodeBench score |
> | -------------- | ------------------- |
> | Base           | 55.76               |
> | STD (32B)      | 58.08               |
> | STD (QWQ)      | 56.88               |
> | STD (235B)     | 58.89               |
> | STD (R1)       | 53.89               |
> | **MoT (ours)** | **61.08**           |
>
> As in the math setting, **MoT again outperforms the best single-teacher STD**, indicating that the method is **not limited to mathematical reasoning** and can also strengthen code reasoning under the same multi-teacher long-CoT setup.
>
> In addition, as reported in our main paper (Table 7), MoT improves performance on a range of **general benchmarks** (e.g., CEVAL, MMLU variants, coding and physics benchmarks) while incurring smaller drops on catastrophic-forgetting–sensitive tasks compared to the best single-teacher STD, further supporting its ability to transfer beyond a single mathematical benchmark.
>
> We agree that a broader sweep over other non-math domains (e.g., large-scale scientific QA) would further strengthen the claim, and we view this as valuable future work. The current results on **code reasoning + general benchmarks**, however, already provide concrete evidence that MoT generalizes beyond competition math.
>
> We are grateful for the reviewer’s detailed and constructive feedback, which we have now updated the corresponding part in the revised version at lines 946–958.
>
> [1] *From Trial-and-Error to Improvement: A Systematic Analysis of LLM Exploration Mechanisms in RLVR*

---

> ### Author Response · Authors · 2025-11-20
> **Author Response to Reviewer HWmF (Part V)**
>
> > **Response to Weakness 3 Part II (Compute–performance trade-off)**
>
> We acknowledge the reviewer’s concern about the trade-off between computational cost and performance. There is indeed a trade-off, and we here discuss the the budgets:
>
> | Method     | # branches | steps / branch | rounds | total branch-steps |
> |-----------|------------|----------------|--------|--------------------|
> | STD       | 1          | 250            | 1      | 250                |
> | **MoT**   | 4          | 50             | 5      | **4 × 50 × 5 = 1000** |
>
> However, MoT is designed for the realistic setting where multiple teachers are available and one wishes to maximally leverage them rather than commit to a single teacher. Moreover, it only takes 6+ GPU hours to fully reproduce the MoT training of QWEN3+BOBA-200 on a single 8xh800 machine and this procedure can be further accelerated by running branches in parallel. We believe this still achieves a very promising effect in a highly efficient way.
>
>
> In addition, to directly address the compute concern, we ran an additional experiment following the RFT-style setting in [1]. For each teacher, we **re-distilled the data** by sampling **4 diverse CoT trajectories per question with high temperature**, yielding **4 × 200 = 800** rationales per teacher. This matches MoT’s total number of distinct CoT sequences (**4 teachers × 200 questions × 1 CoT each = 800**), so the comparison controls for both **total compute** and **the amount of distinct CoT supervision**.
>
> We then performed STD for **1000 steps** on the chosen teacher’s 800 CoTs, matching the *total branch-steps* of MoT:
>
> - MoT: 4 branches × 50 steps/branch × 5 rounds = **1000 branch-steps**
> - RFT-style STD: single teacher, 4 CoTs per question (800 CoTs total), **1000 steps**
>
> We saved a checkpoint every 200 steps and report the best checkpoint. The results on AIME24/25 (Qwen3-8B, BOBA-200) are:
>
> | Method         | AIME24    | AIME25    | AVG       |
> | -------------- | --------- | --------- | --------- |
> | Base           | 75.83     | 67.08     | 71.46     |
> | 4×STD (32B)    | 75.21     | 67.92     | 71.57     |
> | 4×STD (QWQ)    | 76.46     | 69.17     | 72.82     |
> | 4×STD (235B)   | 75.83     | 68.13     | 71.98     |
> | 4×STD (R1)     | 70.42     | 59.58     | 65.14     |
> | **MoT (ours)** | **78.33** | **70.63** | **74.48** |
>
> We found that the previously validated optimal teacher in this setting remains effective on RFT, and its advantage further widens with increased computational resources. Even under this **strictly matched compute and data budget**, the best single-teacher RFT-style STD still lags behind MoT. This suggests that MoT’s gains come not merely from “more compute” but from avoiding the cost and brittleness of teacher selection and unifying multiple teachers’ complementary reasoning abilities, thereby raising the student’s reasoning ceiling.
>
> We appreciate the reviewer’s insightful comment and We have added a clear cost table (as above) and a short discussion of this compute–data–performance trade-off in the revised version accordingly in Section 6.2.
>
> [1] *Scaling Relationship on Learning Mathematical Reasoning with Large Language Models*

---

> ### Author Response · Authors · 2025-11-20
> **Author Response to Reviewer HWmF (Part VI)**
>
> > **Response to Weakness 4 Part I (Baselines from different setups)**
>
>
> We sincerely appreciate the reviewer’s thoughtful concern about the use of baseline numbers taken from prior work with heterogeneous setups. We agree that cross-paper comparisons are not perfectly controlled. Importantly, our **main claims about MoT** are supported by **fully fair, in-house baselines**: for Base / STD / MTD / post-hoc merge / MoT, we use the same base model, same datasets (BOBA-200 / S1K-200), same teacher pool, same training budget, and same evaluation protocol. Under this unified setting, MoT consistently outperforms both the best single-teacher distillation and multi-teacher union across all scales. These are **the most relevant baselines for the real-world scenario** we target (having multiple candidate teachers for the same student and data).
>
>
> For external methods trained on different data, objectives, and pipelines (some SFT, some RL), forcing everything into a single fully unified framework would be both inherently unfair and computationally unaffordable. Instead, we ensure that all compared methods on each block use the same base model, and we evaluate MoT with a standard, widely used protocol: using vLLM to generate answers and extracting the boxed final answer to compute accuracy. Under this common evaluation, with ~200 CoT examples, our MoT models achieve best performance on very competitive reasoning benchmarks, which we view as strong evidence that the proposed training recipe is effective and practically competitive.
>
> ---
>
> We hope these clarifications address your outstanding concerns and are grateful for your helpful guidance. We simply hope that our efforts on this paper can earn your recognition and support. Thank you!
>
> Sincerely,
> *The authors*

---

### Official Review · Reviewer_Came · 2025-10-31

**Soundness:** 2
**Presentation:** 3
**Contribution:** 2
**Rating:** 4
**Confidence:** 3

**Summary:**

This paper tackles the problem of distilling long chain-of-thought (CoT) reasoning from multiple teacher models. The authors start with the key observation that there is no single "best teacher" for all students or datasets. To address this, they propose Merge-of-Thought Distillation (MoT), a lightweight, iterative framework. In each round of MoT, the student model is branched out and finetuned separately on the CoT data from each individual teacher. These branches are then merged back into a single model via weight averaging. This merged model becomes the initialization for the next round. The authors show that this method is highly data-efficient, enabling a Qwen3-14B student trained on only ~200 CoT samples to surpass strong baselines like Deepseek-R1 and Qwen3-32B. The paper also provides analysis suggesting MoT mitigates catastrophic forgetting, is robust to distribution-shifted teachers, and creates a "consensus CoT" in a flatter loss landscape.

**Strengths:**

Strong Motivation & Problem: The paper is motivated by a clear, practical, and important problem. The initial analysis (Fig. 1, Table 1) showing that the optimal teacher varies by student and dataset is a strong justification for moving beyond single-teacher distillation.

Impressive Empirical Results: The performance gains are the primary strength of this paper. The claim that MoT with only 200 samples can elevate a 14B model to outperform a 32B model (Table 3) is a very strong and compelling result.

Data Efficiency: The ability to achieve these results with such a small, 200-sample dataset is highly significant and makes the method practical and accessible.

Thorough Analysis: The authors provide a comprehensive set of experiments, comparing MoT not only to the best single-teacher distillation (STD) but also to naive multi-teacher data mixing (MTD) and one-shot post-hoc merging (Table 2, Table 4). The additional analyses on catastrophic forgetting (Table 5) and the loss landscape (Appendix D) are valuable additions.

**Weaknesses:**

Despite the strong results, the paper's contribution seems to lie more in a successful application of existing techniques than in a novel methodological breakthrough, which makes it borderline.

- Limited Novelty: The core mechanism of MoT (iteratively training separate models on sharded data and averaging their weights) is conceptually almost identical to well-established algorithms like Federated Averaging (FedAvg) in federated learning. While the application (multi-teacher CoT distillation) is new, the method itself (iterative SFT + weight averaging) is not. The paper does not sufficiently differentiate MoT from this and other related model-merging literature.

- Underdeveloped Merging Strategy: The "Merge" in "Merge-of-Thought" is implemented as simple parameter averaging. Given the extensive and recent literature on more sophisticated model merging techniques (e.g., TIES, DARE, task arithmetic), restricting the method to simple averaging feels like a missed opportunity. It is unclear if the gains come from the iterative process alone, or if the merging step itself could be further optimized.

- Unclear Computational Cost: The paper emphasizes data efficiency (200 samples) and total "steps" (250). However, the MoT process involves training K (here, K=4) branches for 50 steps in each of 5 rounds. This totals (4 branches * 50 steps/branch * 5 rounds) = 1000 total branch-steps, which is 4x the computational cost of the 250-step STD/MTD baselines. This significant compute-for-data trade-off is not clearly discussed.

- Indirect Evidence for "Consensus CoT": The claim that MoT "eliminates teacher-specific inductive biases" is very strong. The evidence provided (Fig. 5) shows that MoT has lower confidence on stylistic tokens from one teacher. This is interesting but indirect; it could be interpreted as the averaging process simply "washing out" or "diluting" a specific teacher's signal, rather than intelligently "eliminating" its bias while preserving a core "consensus."

**Questions:**

1. Could the authors explicitly discuss the relationship between MoT and Federated Averaging (FedAvg)? How do they see the conceptual novelty of MoT in light of FedAvg, which also performs iterative, sharded training followed by weight averaging?

2. Why was simple parameter averaging chosen as the only merging strategy? Were more advanced merging techniques explored? Would they be compatible with the iterative MoT framework, and could they potentially offer further gains?

3. Could the authors provide a clear comparison of the total computational cost (e.g., total branch-steps or GPU-hours) for MoT versus the STD and MTD baselines? This would clarify the compute/data/performance trade-off.

---

> ### Author Response · Authors · 2025-11-20
> **Author Response to Reviewer Came (Part I)**
>
> Thank you for your prompt follow-up and for carefully reading our additional experiments! We sincerely hope the following statements can address your concern.
>
> > **Response to Weakness 1 / Question 1 (Novelty vs. FedAvg)**
>
> We thank the reviewer for highlighting the connection to Federated Averaging (FedAvg); this is a very relevant comparison. We fully acknowledge that the mechanical pattern of MoT with local training on shards followed by weight averaging is similar in form to FedAvg, and we do not claim that “iterate SFT + average weights” is a new optimization primitive.
>
> Our contribution is in **how and where this idea is applied**:
>
> 1. **Different goal and setting from FedAvg.**
>    FedAvg addresses *data privacy / distribution* issues: different clients hold **different inputs** (x) and (ideally) the *same* labeling function, and averaging approximates training on the *pooled* data.
>    In contrast, MoT explicitly tackles **multi-teacher long-CoT distillation on the same questions**:
>
>    * all branches see the **same inputs** but **different teacher CoTs**,
>    * these rationales can **disagree token-wise** even when final answers match,
>    * the core problem is to **unify conflicting teacher supervision** and remove teacher-specific biases.
>
>     Mechanically both use averaging, but conceptually MoT is about **building consensus reasoning on shared questions**, not about training on sharded private datasets.
>
> 2. **Iterative merging shown to be effective in long-CoT distillation.**
>     At the same time, we believe that how this FedAvg-like mechanism is motivated and validated in our setting is itself a meaningful contribution. We first revisited revisited one-shot merging and naive multi-teacher union and empirically showed that they do not reliably resolve cross-teacher conflicts. This negative evidence is what motivated us to adopt an iterative merge-of-branches procedure, even though its structure resembles FedAvg. Our experiments then demonstrate that, in the multi-teacher long-CoT distillation setting, this simple procedure is consistently and substantially effective. In this sense, we see our main contribution not as introducing a new averaging operator, but as **identifying and carefully validating a simple, FedAvg-style training recipe that is strongly motivated by our prior analysis and proves highly effective for a new and practically important setting: multi-teacher long-CoT distillation**.

---

> ### Author Response · Authors · 2025-11-20
> **Author Response to Reviewer Came (Part II)**
>
> > **Response to Weakness 2 / Question 2 (Merging strategy: simple averaging vs. advanced merges)**
>
> We sincerely appreciate the reviewer’s thoughtful suggestion to more directly engage with the recent literature on advanced model merging (e.g., TIES, DARE). Our initial choice of **simple parameter averaging** was motivated by two considerations: (i) it is the most widely used and stable merge primitive, making the core behavior of MoT easy to interpret, and (ii) it introduces no extra hyperparameters or filtering heuristics, which is appealing in a delicate long-CoT distillation setting with heterogeneous teachers. That said, we agree that testing more sophisticated strategies is important to understand where the gains are coming from.
>
> During the rebuttal period, we therefore added experiments on **Qwen3-8B + BOBA-200** with both **advanced one-shot merges** and **MoT variants that replace simple averaging with these operators**:
>
> - One-shot TIES merge (TIES)
> - One-shot DARE merge (DARE)
> - MoT(TIES): MoT where the merge step uses TIES instead of averaging
> - MoT(DARE): MoT where the merge step uses DARE instead of averaging
>
> All other settings (teachers, data, schedule, evaluation) are identical. The final AIME AVG is:
>
> | Method                       | AVG   |
> |------------------------------|-------|
> | Base                         | 71.46 |
> | One-shot TIES                | 71.67 |
> | One-shot DARE                | 60.42 |
> | **MoT (TIES)**               | 73.34 |
> | **MoT (DARE)**               | 74.17 |
> | **MoT (simple merge, ours)** | **74.48** |
>
> We also report the behavior across merge rounds:
>
> **MoT(TIES)**
>
> | Round | AIME24 | AIME25 | AVG   |
> |-------|--------|--------|-------|
> | 1     | 73.33  | 66.67  | 70.00 |
> | 2     | 77.50  | 67.50  | 72.50 |
> | 3     | 77.50  | 69.17  | 73.34 |
> | 4     | 74.17  | 65.00  | 69.59 |
> | 5     | 75.00  | 65.83  | 70.42 |
>
> **MoT(DARE)**
>
> | Round | AIME24 | AIME25 | AVG   |
> |-------|--------|--------|-------|
> | 1     | 75.00  | 62.50  | 68.75 |
> | 2     | 76.67  | 65.00  | 70.84 |
> | 3     | 78.33  | 70.00  | 74.17 |
> | 4     | 79.17  | 69.17  | 74.17 |
> | 5     | 73.33  | 65.83  | 69.58 |
>
> These results suggest three points:
>
> 1. **Advanced one-shot merges alone are not sufficient in our setting.**
>    One-shot TIES gives only a very small improvement over the base model, and one-shot DARE leads to a substantial drop in performance. This indicates that more sophisticated methods originally designed for merging models trained on different domains/tasks may be much less directly suitable for **unifying different reasoning paths for the same questions**.
>
> 2. **The iterative MoT process is compatible with and consistently boosts advanced merges.**
>    When we place these same operators(TIES/DARE) **inside the MoT loop**, performance improves substantially, and the DARE's gain of almost **+14 points** is very superising. Algorithmically, this is plausible because one-shot DARE acts on highly conflicting teacher-specific updates all at once, so its aggressive sparsification tends to over-prune useful but partially misaligned reasoning directions and hurts performance. In contrast, **DARE inside MoT sees smaller, progressively more aligned deltas across rounds and branches**, so it behaves like a gradual consensus regularizer that keeps directions repeatedly reinforced by multiple teachers, which explains the large gain when used iteratively. All of this indicates that the key gains indeed come from the iterative merge-and-train procedure.
>
> 3. **Simple averaging still achieves the highest and most robust ceiling.**
>    We found that, unlike MoT based on SIMPLE MERGE, TIES and DARE showed faster convergence in the evaluation of across merge round, but seemed to have a slightly lower upper ceiling and much poorer anti-overfitting ability. A plausible explanation, consistent with our Section 7 discussion, is that simple averaging **does not impose any parameter filtering**, allowing MoT to naturally absorb useful signals even from suboptimal/noisy teachers and thereby achieve a higher reasoning ceiling.
>
> In summary, we chose simple averaging initially for its stability and simplicity, and our new experiments confirm that:
> - more advanced one-shot merges are not enough, even bad in the multi-teacher long-CoT setting;
> - the **iterative MoT framework reliably boosts** whichever merge operator it uses;
> - the simplest merge (plain averaging) inside MoT is surprisingly the **strongest and most robust choice** in this regime.
>
> We have also modified the manuscript to incorporate this thoughtful suggestion, as shown at lines 1000–1024.

---

> ### Author Response · Authors · 2025-11-20
> **Author Response to Reviewer Came (Part III)**
>
> > **Response to Weakness 3 / Question 3 (Computational cost and trade-off)**
>
> We recognize the reviewer’s concern about the trade-off between computational cost and performance. There is indeed a trade-off, and we here discuss the the budgets:
>
> | Method     | # branches | steps / branch | rounds | total branch-steps |
> |-----------|------------|----------------|--------|--------------------|
> | STD       | 1          | 250            | 1      | 250                |
> | **MoT**   | 4          | 50             | 5      | **4 × 50 × 5 = 1000** |
>
> However, MoT is designed for the realistic setting where multiple teachers are available and one wishes to maximally leverage them rather than commit to a single teacher. Moreover, it only takes 6+ GPU hours to fully reproduce the MoT training of QWEN3+BOBA-200 on a single 8xh800 machine and this procedure can be further accelerated by running branches in parallel. We believe this still achieves a very promising effect in a highly efficient way.
>
>
>
>
> In addition, to directly address the compute concern, we ran an additional experiment following the RFT-style setting in [1]. For each teacher, we **re-distilled the data** by sampling **4 diverse CoT trajectories per question with high temperature**, yielding **4 × 200 = 800** rationales per teacher. This matches MoT’s total number of distinct CoT sequences (**4 teachers × 200 questions × 1 CoT each = 800**), so the comparison controls for both **total compute** and **the amount of distinct CoT supervision**.
>
> We then performed STD for **1000 steps** on the chosen teacher’s 800 CoTs, matching the *total branch-steps* of MoT:
>
> - MoT: 4 branches × 50 steps/branch × 5 rounds = **1000 branch-steps**
> - RFT-style STD: single teacher, 4 CoTs per question (800 CoTs total), **1000 steps**
>
> We saved a checkpoint every 200 steps and report the best checkpoint. The results on AIME24/25 (Qwen3-8B, BOBA-200) are:
>
> | Method         | AIME24    | AIME25    | AVG       |
> | -------------- | --------- | --------- | --------- |
> | Base           | 75.83     | 67.08     | 71.46     |
> | 4×STD (32B)    | 75.21     | 67.92     | 71.57     |
> | 4×STD (QWQ)    | 76.46     | 69.17     | 72.82     |
> | 4×STD (235B)   | 75.83     | 68.13     | 71.98     |
> | 4×STD (R1)     | 70.42     | 59.58     | 65.14     |
> | **MoT (ours)** | **78.33** | **70.63** | **74.48** |
>
> We found that the previously validated optimal teacher in this setting remains effective on RFT, and its advantage further widens with increased computational resources. Even under this **strictly matched compute and data budget**, the best single-teacher RFT-style STD still lags behind MoT. This suggests that MoT’s gains come not merely from “more compute” but from avoiding the cost and brittleness of teacher selection and unifying multiple teachers’ complementary reasoning abilities, thereby raising the student’s reasoning ceiling.
>
>
> We appreciate the reviewer’s insightful comment and We have added a clear cost table (as above) and a short discussion of this compute–data–performance trade-off in the revised version accordingly in Section 6.2.
>
> [1] *Scaling Relationship on Learning Mathematical Reasoning with Large Language Models*

---

> ### Author Response · Authors · 2025-11-20
> **Author Response to Reviewer Came (Part IV)**
>
> > **Response to Weakness 4 (Evidence for “consensus CoT” vs. simple washing-out)**
>
> We are grateful for the reviewer’s thoughtful question about whether MoT genuinely captures consensus reasoning rather than merely averaging away differences. We expand on the evidence supporting our claim below:
>
> We agree that Fig. 5 alone is an indirect signal, and we do **not** intend the claim to rest only on a single token-visualization example. Our “consensus CoT” statement is supported by **multiple, complementary pieces of evidence** in Section 8 and the appendices, not just by lower confidence on stylistic tokens.
>
> Concretely, beyond Fig. 5:
>
> * **Behaviorally**, the student-as-teacher experiments in Section 8 show that MoT-trained students produce CoTs that *consistently* yield stronger downstream students than both the Base and Best-STD teachers, suggesting that MoT is not merely diluting signals but producing **more useful and more consistent rationales**.
> * **Optimization-wise**, the loss-landscape and linear mode connectivity probe indicate that MoT converges to **flatter, more stable regions** than MTD, which aligns with the view that cross-teacher conflicts are being mitigated rather than naively averaged away.
> * **Theoretically**, the “two-bonus” decomposition shows that expert-wise preconditioning both **amplifies agreement directions** and **reduces interference on disagreement directions**, providing a mechanism for reinforcing shared reasoning while suppressing conflicting components.
>
> Most importantly, this is reflected in the **significantly strong overall performance**: MoT consistently and significantly **outperforms every single-teacher distillation** on the same data, and surpasses the performance region of baselines that are trained with orders of magnitude more data. In our view, this strong and robust empirical advantage is the clearest evidence that MoT is not merely “washing out” teachers, but is effectively capturing and leveraging a consensus component of their reasoning.
>
>
>
> ---
>
> We humbly seek to ascertain whether our responses adequately address your concerns. We simply hope that our efforts on this paper can earn your recognition and support. Thank you once again!
>
> Sincerely,
> *The authors*

---

### Official Review · Reviewer_6CXT · 2025-11-01

**Soundness:** 3
**Presentation:** 3
**Contribution:** 3
**Rating:** 6
**Confidence:** 4

**Summary:**

This paper introduces Merge-of-Thought Distillation (MoT), a novel framework for distilling long chain-of-thought (CoT) reasoning capabilities from multiple, potentially heterogeneous, teacher LLMs into a single, smaller student model. The core motivation is that there is no single "best teacher" for a given student and dataset, and naive methods like mixing teacher data or one-shot model merging fail to reconcile conflicting supervision signals. MoT addresses this by iteratively alternating between two steps: (1) Teacher-Specific SFT (Branch Training), where the student is fine-tuned on each teacher's reasoning paths, and (2) Weight-Space Merging, where the parameters of these specialized branches are averaged. This process is designed to reinforce consensus reasoning features while suppressing teacher-specific noise and inductive

**Strengths:**

While model merging and CoT distillation are active areas of research, their combination in an iterative, multi-teacher co-distillation framework is novel.
The paper is generally well-written and structured.

**Weaknesses:**

The premise of the paper, "different students have different best teachers," is only derived from two math evaluation datasets. An insufficient number of evaluation datasets may lead to biased conclusions.
The core merging operation is a simple uniform average of parameters. While the results show this is effective, the field of model merging has advanced with more sophisticated techniques. The paper would be strengthened by a direct comparison against a more advanced merging baseline to confirm that the gains are from the iterative process and not just a more clever one-shot merge.

**Questions:**

In evaluation sets outside of mathematical reasoning, does the phenomenon of "Teacher choices are not universal" still exist?
Is the number of teacher models the more, the better?
How does the model perform on the HMMT benchmark?
How were BOBA-200 and S1k-200 sampled? Are there any guiding principles? For reasoning tasks [1], the quality of prompts obtained through random sampling varies significantly.
How does this method perform on other series of models, such as LLaMA?

[1] Reinforcement Learning for Reasoning in Large Language Models with One Training Example

---

> ### Author Response · Authors · 2025-11-20
> **Author Response to Reviewer 6CXT (Part I)**
>
> We express our gratitude for your favorable feedback and valuable insights! We sincerely hope that our response can address your concern.
>
> > **Response to Weakness 1 / Question 1 / Question 3
> (“Teacher choices are not universal” beyond math, and HMMT results)**
>
> We appreciate the reviewer’s concern that our original evidence for “different students have different best teachers” was based on two math benchmarks, and the request to check other domains and HMMT.
>
> ---
>
> ### (1) Evidence beyond math: code reasoning
>
> To test whether “teacher choices are not universal” holds outside pure math, we ran MoT and all STD baselines on a **code reasoning** subset. Concretely, we extracted **178 code-domain examples** from [1] and applied **exactly the same distillation, training, and evaluation pipeline as in the main paper** (same teacher pool, 1Q–multiA CoTs, Qwen3-8B student, vLLM-based evaluation). The LiveCodeBench scores are:
>
> | Method             | LiveCodeBench |
> | ------------------ | ------------- |
> | Base               | 55.76         |
> | STD (32B)          | 58.08         |
> | STD (QWQ)          | 56.88         |
> | STD (235B)         | **58.89**     |
> | STD (R1)           | 53.89         |
> | **MoT (ours)**     | **61.08**     |
>
> On BOBA-200 and S1K-200 (math), the **best single teacher for Qwen3-8B was QWQ** (Table 1 in the paper). On this **code subset**, however, the **best single teacher becomes Qwen3-235B**, while QWQ is no longer the top performer. This cross-domain shift in the “best teacher” provides additional evidence that **teacher choices are not universal**, even for the *same student*, and supports the premise of our work beyond mathematical reasoning. At the same time, **MoT again outperforms all STD variants**, showing that the method itself generalizes to code reasoning.
>
> ---
>
> ### (2) Additional math benchmark: HMMT
>
> We also evaluated Qwen3-8B distilled on **BOBA-200** using MoT and the four STD teachers on the **HMMT** benchmark, to further diversify math evaluation. The results are:
>
> | Method             | HMMT score |
> | ------------------ | ---------- |
> | Base               | 38.33      |
> | STD (32B)          | 43.33      |
> | STD (QWQ)          | **48.33**  |
> | STD (235B)         | 45.83      |
> | STD (R1)           | 40.83      |
> | **MoT (ours)**     | **52.50**  |
>
> Here, **QWQ remains the best single teacher**, consistent with Table 1 in our main paper for this student/dataset setting, which supports the stability of our teacher-selection analysis under this fixed configuration. Importantly, **MoT still achieves the highest score**, improving over the best STD and reinforcing that MoT effectively **unifies multiple teachers’ reasoning abilities** and raises the student’s reasoning ceiling.
>
> ---
>
> ### (3) Summary
>
> Across different evaluation domains(AIME24/25, HMMT and code reasoning domain), we observe that the **identity of the “best” teacher changes** with the dataset/domain (e.g., QWQ vs Qwen3-235B), supporting our claim that **teacher choices are not universal**, and MoT consistently outperforms all single-teacher STDs, confirming the effectiveness of our multi-teacher consensus distillation.
>
> We agree that a broader sweep (e.g., more scientific QA datasets) would further strengthen this claim and see that as valuable future work, but the new code and HMMT experiments already provide additional, concrete evidence beyond the two original math datasets.
>
> We are grateful for the reviewer’s detailed and constructive feedback, which we have now updated the corresponding part in the revised version at lines 946–998.
>
> [1] *From Trial-and-Error to Improvement: A Systematic Analysis of LLM Exploration Mechanisms in RLVR*

---

> ### Author Response · Authors · 2025-11-20
> **Author Response to Reviewer 6CXT (Part II)**
>
> > **Response to Weakness 2 “need stronger comparison with advanced merging methods”**
>
> Thank you for the thoughtful suggestion. To more fully address this concern, we expanded our study to incorporate several advanced model-merging baselines and integrated them into the MoT pipeline.
>
> On **Qwen3-8B + BOBA-200**, we evaluated:
>
> - **One-shot TIES merge** (TIES)
> - **One-shot DARE merge** (DARE)
> - **MoT variants that replace simple averaging with TIES or DARE in the merge step**
>
> All other settings (teachers, data, schedule, evaluation) are identical. The final AIME AVG is:
>
> | Method                       | AVG   |
> |------------------------------|-------|
> | Base                         | 71.46 |
> | One-shot TIES                | 71.67 |
> | One-shot DARE                | 60.42 |
> | **MoT (TIES)**               | 73.34 |
> | **MoT (DARE)**               | 74.17 |
> | **MoT (simple merge, ours)** | **74.48** |
>
> We also report the **per-round behavior** of MoT(TIES) and MoT(DARE):
>
> **MoT(TIES)**
>
> | Round | AIME24 | AIME25 | AVG   |
> |-------|--------|--------|-------|
> | 1     | 73.33  | 66.67  | 70.00 |
> | 2     | 77.50  | 67.50  | 72.50 |
> | 3     | 77.50  | 69.17  | 73.34 |
> | 4     | 74.17  | 65.00  | 69.59 |
> | 5     | 75.00  | 65.83  | 70.42 |
>
> **MoT(DARE)**
>
> | Round | AIME24 | AIME25 | AVG   |
> |-------|--------|--------|-------|
> | 1     | 75.00  | 62.50  | 68.75 |
> | 2     | 76.67  | 65.00  | 70.84 |
> | 3     | 78.33  | 70.00  | 74.17 |
> | 4     | 79.17  | 69.17  | 74.17 |
> | 5     | 73.33  | 65.83  | 69.58 |
>
> From these results we observe:
>
> 1. **Advanced one-shot merges alone are not sufficient in our setting.**
>    One-shot TIES only slightly improves over the base model, and one-shot DARE even causes a **severe performance drop**. This indicates that more sophisticated methods originally designed for merging models trained on different domains/tasks may be much less directly suitable for **unifying different reasoning paths for the same questions**.
>
> 2. **The iterative MoT process consistently adds value on top of advanced merges.**
>    When we place these same operators(TIES/DARE) **inside the MoT loop**, performance improves substantially, and the DARE's gain of almost **+14 points** is very superising. Algorithmically, this is plausible because one-shot DARE acts on highly conflicting teacher-specific updates all at once, so its aggressive sparsification tends to over-prune useful but partially misaligned reasoning directions and hurts performance. In contrast, **DARE inside MoT sees smaller, progressively more aligned deltas across rounds and branches**, so it behaves like a gradual consensus regularizer that keeps directions repeatedly reinforced by multiple teachers, which explains the large gain when used iteratively. All of this indicates that the key gains indeed come from the iterative merge-and-train procedure.
>
> 3. **Simple averaging still achieves the highest and most robust ceiling.**
>    We found that, unlike MoT based on SIMPLE MERGE, TIES and DARE showed faster convergence in the evaluation of across merge round, but seemed to have a slightly lower upper ceiling and much poorer anti-overfitting ability. A plausible explanation, consistent with our Section 7 discussion, is that simple averaging **does not impose any parameter filtering**, allowing MoT to naturally absorb useful signals even from suboptimal/noisy teachers and thereby achieve a higher reasoning ceiling.
>
> Overall, these experiments confirm that:
> - more advanced one-shot merges are not enough, even bad in the multi-teacher long-CoT setting;
> - the **iterative MoT framework reliably boosts** whichever merge operator it uses;
> - the simplest merge (plain averaging) inside MoT is surprisingly the **strongest and most robust choice** in this regime.
>
> We have also modified the manuscript to incorporate this thoughtful suggestion, as shown at lines 1000–1024.

---

> ### Author Response · Authors · 2025-11-20
> **Author Response to Reviewer 6CXT (Part III)**
>
> > **Response to Question 2 (Is “more teachers” always better?)**
>
> We thank the reviewer for highlighting this question. In our current setting, adding more teachers is indeed consistently beneficial, even when the extra teacher is relatively noisy. We have also partially discussed this in Section 7.
>
> Concretely, on BOBA-200 with Qwen3-8B, we start from the best single teacher (by STD performance), then progressively add the second-best, third-best, and finally the noisy teacher R1 into the MoT pool, keeping the MoT configuration fixed. The AIME24/25 results are:
>
> | # Teachers in MoT | AIME24 | AIME25 | AVG   |
> | ----------------- | ------ | ------ | ----- |
> | 1 (best teacher)  | 76.25  | 67.50  | 71.88 |
> | 2                 | 77.50  | 68.33  | 72.92 |
> | 3                 | 78.13  | 69.17  | 73.65 |
> | 4 (with  R1)      | 78.33  | 70.63  | 74.48 |
>
> We observe a monotonic improvement in AIME AVG as we add more teachers, and performance continues to improve even after including the distribution-shifted / noisy teacher R1. This supports the view that, in our experiment regime, MoT can effectively extract complementary signals from additional teachers and is robust enough to benefit from them.
>
> That said, we do not claim this will hold for an arbitrarily large pool of very low-quality or adversarial teachers; in such extreme cases we would expect diminishing returns and a need for stronger filtering or weighting. Systematically studying how performance scales with larger and more heterogeneous teacher sets is an interesting direction for future work.
>
> We appreciate the reviewer’s insightful comment and have revised the paper accordingly at lines 1024–1055.
>
> > **Response to Question 4 (Sampling of BOBA-200 and S1K-200, and guiding principles)**
>
> We appreciate the reviewer’s concern about sampling quality for reasoning tasks.
>
> * **BOBA-200.**
>   For BOBA-200, we simply use the **default 200 problems** provided by the official BOBA release, which are themselves obtained by **random sampling** from the full BOBA benchmark. We did **not** apply any additional filtering or manual selection, to avoid introducing cherry-picking or selection bias.
>
> * **S1K-200.**
>   For S1K, some items are proof-style questions without a clearly verifiable final answer (e.g., no boxed numeric answer). Since our distillation and evaluation pipeline requires **automatic answer checking**, our guiding principle was:
>
>   1. First, remove all proof problems without a verifiable final answer (about 200 items).
>   2. Then, from the remaining problems that do have a boxed / checkable answer, we uniformly sample 200 prompts at random to form S1K-200.
>
> We fully agree with [1] that random sampling can lead to variable prompt difficulty. **We have discussed it together with our sampling guiding principles in the revised version at lines 264–274**. So, in both cases, the guiding principles were:
>
> * keep the process **as random as possible** within the constraints of **automatic answer verification**,
> * avoid any manual cherry-picking, and ensure the subsets are reproducible and suitable for CoT distillation.
> * Ensure that all comparative experiments of the baseline of MoT and all STDS and MTD are completed based on a batch of exactly the same prompts, and minimize the risk that our conclusions hinge on a specific sample.
>
> [1] Reinforcement Learning for Reasoning in Large Language Models with One Training Example
>
>
>
> > **Response to Question 5 (Other model series such as LLaMA)**
>
> We acknowledge the reviewer’s insightful concern regarding the external validity of our results beyond the Qwen family. To test this, we replicated our BOBA-200 distillation setup on Llama-3.1-8B-Instruct model, using the same teacher pool and the same MoT pipeline. The results are:
>
> | Method            | Math500 score |
> |-------------------|---------------|
> | Base              | 49.65         |
> | STD (Qwen3-32B)   | 55.45         |
> | STD (QWQ)         | 55.80         |
> | STD (Qwen3-235B)  | 57.45         |
> | STD (Deepseek-R1) | 53.80         |
> | **MoT (ours)**    | **62.65**     |
>
> As with the Qwen experiments, MoT substantially improves the Llama student and clearly outperforms the best single-teacher STD. This supports our claim that MoT is **not tied to Qwen-specific design choices**, but instead is a lightweight, architecture-agnostic procedure for multi-teacher long-CoT distillation.
>
> We sincerely thank the reviewer for this constructive suggestion; we have incorporated the change into the manuscript at lines 900–908.
>
> ---
>
> We hope these clarifications address your outstanding concerns and are grateful for your helpful guidance. We very much hope that our revisions and additional efforts in this paper can earn your recognition and support. Thank you once again!
>
> Sincerely,
> *The authors*

---

> > ### Comment · Reviewer_6CXT · 2025-11-26
> >
> > Thank you for your detailed reply. I will keep this positive score.

---

> > > ### Author Response · Authors · 2025-11-26
> > > **Thank you for your response**
> > >
> > > Dear Reviewer,
> > >
> > > Thank you very much for your constructive feedback and for taking the time to carefully review our work! We sincerely appreciate your thoughtful comments and suggestions, which were highly valuable in helping us refine the paper and clarify several important points. Your comments showed us what needed fixing and pushed us to add the right experiments. The paper is better because of your review.
> > >
> > > If any additional questions arise or if further discussion would be useful, we would be more than happy to continue the conversation. Thank you again for your constructive engagement with our submission.
> > >
> > > Thanks again,
> > >
> > > The authors

---

### Official Review · Reviewer_9K2Y · 2025-11-01

**Soundness:** 2
**Presentation:** 3
**Contribution:** 3
**Rating:** 4
**Confidence:** 3

**Summary:**

The paper proposes Merge-of-Thought Distillation (MoT), an iterative multi-teacher CoT
distillation framework that alternates between (a) teacher-specific supervised fine-tuning
branches and (b) weight-space averaging of the resulting student variants. The aim is to fuse
complementary reasoning signals from heterogeneous teachers while suppressing teacher-
specific noise and conflicts. Empirically, the authors show that the “best” single teacher depends
on both the student and the dataset, and that naïve multi-teacher data union or one-shot post-hoc
merging does not reliably resolve cross-teacher conflicts. MoT is reported to outperform single-
teacher and naïve multi-teacher baselines on AIME24/25 using only ~200 CoT exemplars, with
additional evidence of smaller catastrophic-forgetting drops and gains on non-math reasoning
benchmarks.

**Strengths:**

1. The branch-train + weight-merge loop is easy to implement; it sidesteps brittle manual teacher selection.
2. Demonstrates competitive AIME improvements using only ~200 CoT samples, even versus larger models.
3. MoT not only strengthens general reasoning but also helps mitigate catastrophic forgetting compared to best STD

**Weaknesses:**

1. Experiments focus primarily on the Qwen family and math-centric CoT, with limited  coverage of other backbones or domains. This narrows external validity. The paper itself  notes comparisons primarily around Qwen bases.
2. Several baseline results are taken from papers with different setups, as the original  code/models are unavailable. This weakens the claim of direct superiority.
3. MoT requires training multiple branches across several rounds, making it more compute-intensive than single-teacher distillation, while often underperforming STD in  the first round before eventual gains appear.

**Questions:**

1. How sensitive are MoT gains to random seeds across both branch training and merging rounds, when using a fixed early-stopping rule rather than “best of” selection? Please report mean±std over multiple independent runs.
2. Have you tested LLaMA, Mixtral, or Mistral-based models? Are MoT gains specific to Qwen?
3. Have you tested MoT on other domains (e.g., code reasoning or scientific QA)? If not, what are the expected limitations?
4. How does MoT behave when one teacher is adversarial or noisy? Is the method robust to such imbalance?

---

> ### Author Response · Authors · 2025-11-20
> **Author Response to Reviewer 9K2Y (Part I)**
>
> We would like to thank you for your valuable feedback and far-sighted insights! We hope the following statements will provide you gain an in-depth understanding of our MoT method, thereby dispelling your concerns.
>
> > **Response to Weakness1 & Question 2 (External validity beyond Qwen / other backbones)**
>
> We acknowledge the reviewer’s insightful concern regarding the external validity of our results beyond the Qwen family, and that it is important to verify whether MoT’s gains are specific to those backbones. During the rebuttal period, we therefore replicated our BOBA-200 setup on a *different* architecture family, **Llama-3.1-8B Instruct**. The results are:
>
> | Method             | Math500 score |
> | ------------------ | ------------- |
> | Base               | 49.65         |
> | STD (Qwen3-32B)    | 55.45         |
> | STD (QWQ)          | 55.80         |
> | STD (Qwen3-235B)   | 57.45         |
> | STD (Deepseek-R1)  | 53.80         |
> | **MoT (ours)**     | **62.65**     |
>
> The observations are consistent with our Qwen results: MoT significantly improves Llama 3.1 8B Instruct and still provides a sizable margin over the **best** single-teacher distillation. This supports our claim that MoT is **not tied to Qwen-specific design choices**; it is a lightweight, architecture-agnostic training procedure.
>
> We sincerely thank the reviewer for this constructive suggestion; we have incorporated the change into the manuscript at lines 900–908.
>
>
>
> > **Response to Weakness 2 (Baselines from different setups)**
>
> We appreciate the reviewer’s concern about the use of baseline numbers taken from prior work with heterogeneous setups. We agree that cross-paper comparisons are not perfectly controlled. Importantly, our **main claims about MoT** are supported by **fully fair, in-house baselines**: for Base / STD / MTD / post-hoc merge / MoT, we use the same base model, same datasets (BOBA-200 / S1K-200), same teacher pool, same training budget, and same evaluation protocol. Under this unified setting, MoT consistently outperforms both the best single-teacher distillation and multi-teacher union across all scales. These are **the most relevant baselines for the real-world scenario** we target (having multiple candidate teachers for the same student and data).
>
>
> For external methods trained on different data, objectives, and pipelines (some SFT, some RL), forcing everything into a single fully unified framework would be both inherently unfair and computationally unaffordable. Instead, we ensure that all compared methods on each block use the same base model, and we evaluate MoT with a standard, widely used protocol: using vLLM to generate answers and extracting the boxed final answer to compute accuracy. Under this common evaluation, with ~200 CoT examples, our MoT models achieve best performance on very competitive reasoning benchmarks, which we view as strong evidence that the proposed training recipe is effective and practically competitive.

---

> ### Author Response · Authors · 2025-11-20
> **Author Response to Reviewer 9K2Y (Part II)**
>
> > **Response to Weakness 3 (Compute cost and training multiple branches / rounds)**
>
> We recognize the reviewer’s observation that MoT is more compute-intensive than single-teacher distillation and may underperform STD in early rounds. There is indeed a trade-off between compute and performance, and MoT is designed for the realistic setting where multiple teachers are available and one wishes to maximally leverage them rather than commit to a single teacher. Moreover, it only takes 6+ GPU hours to fully reproduce the MoT training of QWEN3+BOBA-200 on a single 8xh800 machine and this procedure can be further accelerated by running branches in parallel. We believe this still achieves a very promising effect in a highly efficient way.
>
> In addition, to directly address the compute concern, we ran an additional experiment following the RFT-style setting in [1]. For each teacher, we **re-distilled the data** by sampling **4 diverse CoT trajectories per question with high temperature**, yielding **4 × 200 = 800** rationales per teacher. This matches MoT’s total number of distinct CoT sequences (**4 teachers × 200 questions × 1 CoT each = 800**), so the comparison controls for both **total compute** and **the amount of distinct CoT supervision**.
>
> We then performed STD for **1000 steps** on the chosen teacher’s 800 CoTs, matching the *total branch-steps* of MoT:
>
> - MoT: 4 branches × 50 steps/branch × 5 rounds = **1000 branch-steps**
> - RFT-style STD: single teacher, 4 CoTs per question (800 CoTs total), **1000 steps**
>
> We saved a checkpoint every 200 steps and report the best checkpoint. The results on AIME24/25 (Qwen3-8B, BOBA-200) are:
>
> | Method         | AIME24    | AIME25    | AVG       |
> | -------------- | --------- | --------- | --------- |
> | Base           | 75.83     | 67.08     | 71.46     |
> | 4×STD (32B)    | 75.21     | 67.92     | 71.57     |
> | 4×STD (QWQ)    | 76.46     | 69.17     | 72.82     |
> | 4×STD (235B)   | 75.83     | 68.13     | 71.98     |
> | 4×STD (R1)     | 70.42     | 59.58     | 65.14     |
> | **MoT (ours)** | **78.33** | **70.63** | **74.48** |
>
> We found that the previously validated optimal teacher in this setting remains effective on RFT, and its advantage further widens with increased computational resources. Even under this **strictly matched compute and data budget**, the best single-teacher RFT-style STD still lags behind MoT. This suggests that MoT’s gains come not merely from “more compute” but from avoiding the cost and brittleness of teacher selection and unifying multiple teachers’ complementary reasoning abilities, thereby raising the student’s reasoning ceiling.
>
> We appreciate the reviewer’s insightful comment and We have added a clear cost table (as above) and a short discussion of this compute–data–performance trade-off in the revised version accordingly in Section 6.2.
>
> [1] *Scaling Relationship on Learning Mathematical Reasoning with Large Language Models*
>
>
>
> > **Response to Question 1 (Seed sensitivity and fixed early stopping)**
>
> Thank you for the careful question about robustness to random seeds and early stopping. We re-ran **Qwen3-8B + BOBA-200** with **5 independent seeds** for MoT and each STD baseline, and report mean ± std. In addition, we implemented a fixed validation-based early-stopping rule for MoT: we split 10% of the original training set as a held-out validation set for early-stopping, and report performance of final checkpoint.
>
> We keep the same 5-round schedule and compute **AIME AVG = (AIME24 + AIME25)/2** at each round:
>
> | Merge round  | AIME AVG (mean ± std) |
> | -----------  | --------------------- |
> | 1            | 70.17 ± 0.63          |
> | 2            | 72.37 ± 0.65          |
> | 3            | 73.38 ± 0.09          |
> | 4            | **74.89 ± 1.05**      |
> | 5            | 72.75 ± 0.81          |
> |early-stopping| 73.46 ± 0.31          |
>
> This validation-based early stopping yields slightly lower AVG than always taking round 4 (as expected, since the train set is smaller), but still remains strong and better than any STD/MTD baseline, and it also better than the same final checkpoint of MoT without early-stopping, showing that MoT works well under a fixed early-stopping rule that does not use the “best of” selection.
>
> Thank you for this helpful suggestion. We have updated the corresponding part of the manuscript at lines 908–946 to reflect this change.

---

> ### Author Response · Authors · 2025-11-20
> **Author Response to Reviewer 9K2Y (Part III)**
>
> > **Response to Question 3 (Other domains)**
>
> The reviewer’s question about extending MoT beyond mathematical reasoning is well-taken. To directly examine this, we ran MoT on a code-reasoning domain using 178 code-domain examples from [1], and applied exactly the same distillation, training, and evaluation pipeline as in the main paper (same teacher pool, same 1Q–multiA CoT collection, same MoT procedure). We evaluated on LiveCodeBench:
>
> | Method             | LiveCodeBench score |
> | ------------------ | ------------------- |
> | Base               | 55.76               |
> | STD (32B)          | 58.08               |
> | STD (QWQ)          | 56.88               |
> | STD (235B)         | 58.89               |
> | STD (R1)           | 53.89               |
> | **MoT (ours)**     | **61.08**           |
>
> As in the math setting, **MoT again outperforms the best single-teacher STD**, suggesting that MoT is **not restricted to mathematical reasoning** and can also improve code reasoning under the same multi-teacher long-CoT setup.
>
> In addition, as reported in our main paper (Table 7), MoT improves performance on a range of **general benchmarks** (e.g., CEVAL, MMLU variants, coding and physics benchmarks) while incurring smaller drops on catastrophic-forgetting–sensitive tasks compared to the best single-teacher STD, further supporting its ability to transfer beyond a single mathematical benchmark.
>
> We agree that a broader sweep over other non-math domains (e.g., large-scale scientific QA) would further strengthen the claim, and we view this as valuable future work. The current results on **code reasoning + general benchmarks**, however, already provide concrete evidence that MoT generalizes beyond competition math.
>
> We are grateful for the reviewer’s detailed and constructive feedback, which we have now updated the corresponding part in the revised version at lines 946–958.
>
> [1] *From Trial-and-Error to Improvement: A Systematic Analysis of LLM Exploration Mechanisms in RLVR*
>
>
>
> > **Response to Question 4 (Noisy / adversarial teachers and robustness)**
>
> The reviewer’s question about robustness under noisy or adversarial teacher conditions touches on an important practical consideration, and we fully agree that such robustness is essential for real-world multi-teacher training scenarios. Our paper already discuesses this in Section 7 via the Deepseek-R1 (R1) case, which behaves like a distribution-shifted / poor teacher for Qwen students. As shown in Table 4, STD(R1) clearly underperforms other teachers. However, when we include R1 in the MoT teacher pool and then ablate it (MoT vs. MoT–noR1), Table 8 shows that **removing R1 consistently *reduces* MoT performance**. Figure 4 further indicates that MoT with R1 reaches a **higher performance ceiling and less post-peak degradation**. This suggests that MoT can still extract complementary useful signals from a noisy/shifted teacher while mitigating its negative effects.
>
>
> ---
>
> We humbly seek to ascertain whether our responses adequately address your concerns. We would be extremely grateful if you would reconsider your scores to support our work. Thank you once again!
>
>
> Sincerely,
> *The authors*

---

### Author Response · Authors · 2025-11-30
**Summary of Responses to Each Reviewer**

Dear Reviewers, Area Chairs, and Program Chairs,

We would like to express our sincere gratitude for the time, care, and expertise you devoted to evaluating our submission. Your constructive feedback greatly strengthened the clarity, rigor, and breadth of this work. We deeply appreciate the thoughtful effort each of you invested in the review and discussion process.

We summarize our responses to each reviewer as below:

* **External validity across backbones and domains.**
  We extended experiments from Qwen to **Llama-3.1-8B-Instruct**, and from competition math to **code reasoning** and an additional math benchmark (HMMT). In all cases, MoT consistently outperforms the best single-teacher distillation, showing that MoT is **architecture-agnostic** and generalizes beyond pure math.
  *(To Reviewers 9K2Y, 6CXT, HWmF)*

* **Clarifying novelty and relation to FedAvg / model merging.**
  We explicitly discussed MoT’s relationship to FedAvg and prior merging methods, and clarified that our contribution is not just a new optimization primitive but a **carefully motivated and validated training recipe** for the novel setting of **multi-teacher long-CoT distillation on the same questions**, where teachers provide heterogeneous, often conflicting rationales.
  *(To Reviewers Came, HWmF)*

* **Advanced merging baselines and where MoT’s gains come from.**
  We added experiments with TIES and DARE both as one-shot merges and as drop-in replacements for MoT’s merge step. One-shot TIES/DARE are weak or even harmful in our setting, while MoT(TIES) and MoT(DARE) perform strongly, and simple averaging inside MoT remains the best and most robust variant. This shows that the key gains come from the **iterative merge-and-train process** proposed by our MoT, not from a special merge operator.
  *(To Reviewers 6CXT, Came, HWmF)*

* **Compute–data–performance trade-off.**
  We added a **clear cost table** and reported **total GPU hours** (only ≈6h for Qwen3 + BOBA-200 on 8×H800). We further introduced a **compute- and data-matched RFT-style STD baseline** and showed that MoT still achieves higher accuracy, indicating that MoT’s gains are not simply due to using more compute.
  *(To Reviewers 9K2Y, Came, HWmF)*

* **Robustness to random seeds, early stopping, and teacher count.**
  We ran 5-seed experiments with a fixed validation-based early stopping rule, showing MoT still outperforms all STD/MTD baselines without “best-of-round” selection. We also performed a teacher-pool ablation, adding teachers one by one, and observed **monotonic improvements** as more teachers are added, indicating robustness and the ability to exploit complementary teachers.
  *(To Reviewers 9K2Y, 6CXT, HWmF)*

* **Baselines, fairness, and consensus behavior.**
  We clarified that our main claims are supported by **fully controlled in-house baselines** (Base / STD / MTD / post-hoc merge / MoT) under identical models, prompts, budgets, and evaluation protocols. For external numbers from prior work, we explained why a perfectly unified pipeline is not feasible and how we standardize evaluation. We also discussed why MoT **extracts consensus reasoning** rather than merely washing out signals, supported by student-as-teacher experiments, loss-landscape analysis and the “two-bonus” decomposition.
  *(To Reviewers 9K2Y, Came, HWmF)*

* **Data construction and sampling principles.**
  We detailed how **BOBA-200** (official 200 problems) and **S1K-200** (randomly sampled from items with verifiable final answers) are constructed, emphasizing **no cherry-picking** and that all methods compared on a given setting share **exactly the same prompts**, in line with concerns about sampling bias for reasoning tasks.
  *(To Reviewers 6CXT)*

**We have incorporated responses to all reviewers into the latest PDF version, with each modification highlighted in blue throughout the paper.**

Once again, we are truly grateful for your thoughtful comments and the time you invested in improving this work. Thank you for helping us make the paper stronger, clearer, and more broadly relevant.

Best regards,

The authors

---

### Meta-Review · Area_Chair_AiYn · 2026-01-06

**Summary:**

The reviewers raised a mix of concerns and positive feedback that influenced the suggested decision. While the empirical results were strong, particularly in terms of data efficiency and the ability to outperform larger models with limited data, some reviewers questioned the novelty of the core method. They pointed out that the iterative branching and weight averaging approach is conceptually similar to existing techniques like Federated Averaging (FedAvg) and didn't sufficiently differentiate itself from other model-merging strategies. There was also concern about the simplicity of the merging strategy, with some reviewers suggesting that more advanced methods like TIES and DARE should have been explored. Additionally, while the method performed well on mathematical benchmarks, concerns about its generalization to non-mathematical tasks remained, with insufficient demonstration of its ability to handle diverse domains. The trade-off between computational cost and performance gains was also not fully addressed. Despite these concerns, Reviewer 6CXT gave a positive evaluation, acknowledging the strength of the paper's motivation, comprehensive experiments, and data efficiency. However, overall, the combination of concerns about novelty, generalization, and missing advanced comparisons led to a marginal rejection recommendation.

**Reviewer Concerns:**

Addressed:

The authors expanded their experiments to include models like Llama-3.1-8B-Instruct and benchmarks like code reasoning, showing that MoT's improvements are not restricted to specific model architectures or domains.

The authors clarified their decision to use simple parameter averaging as a merging strategy, citing its stability and suitability for this context. They also tested more advanced methods like TIES and DARE, showing that MoT outperforms them due to its iterative process.

The authors provided a clear comparison of the compute costs of MoT and standard distillation methods, showing that MoT achieves substantial performance gains without excessive compute overhead.

The authors provided additional evidence that MoT is not simply "washing out" teacher signals but is effectively capturing consensus reasoning by using a student-as-teacher experiment, loss-landscape analysis, and theoretical justifications.

Still outstanding:

While MoT uses a uniform approach to teacher selection, some reviewers raised concerns about its inability to dynamically assess teacher quality, especially in larger, heterogeneous teacher pools.

Reviewers highlighted that the core MoT method (branch training followed by weight averaging) is conceptually similar to Federated Averaging (FedAvg) and other model-merging techniques. The novelty of MoT lies more in its specific application to multi-teacher CoT distillation, but this point was not always made clear in the initial version of the paper.

While MoT was shown to work beyond mathematical reasoning tasks, some reviewers questioned whether the framework would generalize well to larger-scale scientific QA tasks or other domains not covered in the current experiments.

**Reviewer Scores:**

Reviewer 9K2Y: This reviewer had concerns about the external validity and merging strategies but appreciated the clarity of the rebuttal. They likely would have slightly improved their score from 4 to 5 after the rebuttal, but the concerns about dynamic teacher assessment and novelty of the method may have kept them from fully supporting acceptance.

Reviewer 6CXT: Reviewer 6CXT gave a positive score, noting the empirical results and the strong motivation of the paper. The rebuttal helped address concerns about external validity, demonstrating MoT's generalization beyond math-based benchmarks. However, the reviewer still had concerns about the novelty of the method and its comparison to existing techniques like Federated Averaging. Given the improvements in clarity and the additional experiments, the reviewer might have been convinced to maintain their positive view of the paper, potentially keeping their score at 6.

Reviewer Came: This reviewer raised questions about the novelty and computational cost trade-off. While the authors provided a detailed rebuttal, this reviewer still had doubts about the innovation of the approach and the computational efficiency. They would have likely kept their score at 4, but would not have been completely opposed to acceptance.

Reviewer HWmF: This reviewer had strong concerns about the method's novelty and computational cost. Despite the rebuttal addressing many points, they still felt that the core methodology lacked innovation and that the results could be interpreted as an application of existing techniques. Their score would likely remain 4.

---

### Decision · Program_Chairs · 2026-01-26

Reject